# Junior scientists spotlight social bonds in seminars for diversity, equity, and inclusion in STEM

Evan A. Boyle[1]*, Gabriela Goldberg[1], Jonathan C. Schmok[1], Jillybeth Burgado[2], Fabiana Izidro Layng[3], Hannah A. Grunwald[4], Kylie M. Balotin[5], Michael S. Cuoco[6], Keng-Chi Chang[7], Gertrude Ecklu-Mensah[8], Aleena K. S. Arakaki[9], Noorsher Ahmed[1], Ximena Garcia Arceo[10], Pratibha Jagannatha[1], Jonathan Pekar[11], Mallika Iyer[12], DASL Alliance[¶], Gene W. Yeo[1,13,14]*

1 Department of Cellular & Molecular Medicine, University of California San Diego, La Jolla, CA, United States of America, 2 Molecular Neurobiology Laboratory, The Salk Institute for Biological Studies, San Diego, CA, United States of America, 3 Cancer Center, Sanford Burnham Prebys Medical Discovery Institute, San Diego, CA, United States of America, 4 Department of Genetics, Harvard Medical School, Boston, MA, United States of America, 5 Department of Biomedical Engineering, Vanderbilt University, Nashville, TN, United States of America, 6 Laboratory of Genetics, The Salk Institute for Biological Studies, San Diego, CA, United States of America, 7 Department of Political Science, University of California San Diego, La Jolla, CA, United States of America, 8 Department of Pediatrics, University of California San Diego, La Jolla, CA, United States of America, 9 Human Biology Division, Fred Hutchinson Cancer Research Center, Seattle, Washington, United States of America, 10 Department of Chemistry and Biochemistry, University of California San Diego, La Jolla, CA, United States of America, 11 Department of Biomedical Informatics, University of California San Diego, La Jolla, CA, United States of America, 12 Graduate School of Biomedical Sciences, Sanford Burnham Prebys Medical Discovery Institute, San Diego, CA, United States of America, 13 Stem Cell Program, University of California San Diego, La Jolla, CA, United States of America, 14 Institute for Genomic Medicine, University of California San Diego, La Jolla, CA, United States of America

¶ Membership of the DASL Alliance is listed in the Acknowledgments.
* eboyle@ucsd.edu (EAB); geneyeo@ucsd.edu (GWY)

**Data Availability Statement:** Topic mention search queries and counts per speaker are available in the Supplementary Data. All speaker profiles and self-

## Abstract

Disparities for women and minorities in science, technology, engineering, and math (STEM) careers have continued even amidst mounting evidence for the superior performance of diverse workforces. In response, we launched the Diversity and Science Lecture series, a cross-institutional platform where junior life scientists present their research and comment on diversity, equity, and inclusion in STEM. We characterize speaker representation from 79 profiles and investigate topic noteworthiness via quantitative content analysis of talk transcripts. Nearly every speaker discussed interpersonal support, and three-fifths of speakers commented on race or ethnicity. Other topics, such as sexual and gender minority identity, were less frequently addressed but highly salient to the speakers who mentioned them. We found that significantly co-occurring topics reflected not only conceptual similarity, such as terms for racial identities, but also intersectional significance, such as identifying as a Latina/Hispanic woman or Asian immigrant, and interactions between concerns and identities, including the heightened value of friendship to the LGBTQ community, which we reproduce using transcripts from an independent seminar series. Our approach to scholar profiles and talk transcripts serves as an example for transmuting hundreds of hours of

identification data are available on the DASL website: https://www.ucsddasl.com.

**Funding:** DASL is funded in part by a grant from the Chan-Zuckerberg Initiative. E.A.B. acknowledges funding from the Helen Hay Whitney Foundation. G.W.Y. is an Allen Distinguished Investigator, a Paul G. Allen Frontiers Group advised program of the Paul G. Allen Family Foundation. The funders had no role in study design, data collection and analysis, decision to publish, or preparation of the manuscript.

**Competing interests:** The authors have declared that no competing interests exist.

scholarly discourse into rich datasets that can power computational audits of speaker diversity and illuminate speakers' personal and professional priorities.

## Introduction

Following spring of 2020, broader recognition of widespread social injustice spurred advocacy for diversity, equity, and inclusion (DEI) in STEM. Several teams of junior scientists have published recommendations on faculty hiring, grant review, and diversity initiatives [1–4]. Irrespective of the disparity targeted, guidelines for DEI build on a common foundation: listening to members of historically excluded groups share their experience working in STEM.

In June of 2020, we founded the Diversity and Science Lecture series (DASL), a platform where junior life scientists in San Diego and beyond can share their research and comment on DEI in STEM (Fig 1A). DASL features integrated presentations on speakers' personal backgrounds, their scientific progress, and their advice for navigating a scientific career. Each academic quarter, executive team members recruit speakers and schedule times for dry-run practice sessions and formal seminars. Each seminar consists of either two 15–20 minute talks from trainee life scientists or a 1 hour talk from an early career life scientist or social science expert.

Other prominent virtual seminar series launched in 2020 commonly hewed close to existing scientific communities [5–7]. By contrast, DASL is accessible to a broad cross-section of life scientists. Speakers are encouraged to speak freely about issues important to them and deliver a clear message advising their peers on how to advance DEI in STEM. In June of 2021, DASL concluded a year of weekly sessions and has featured over 100 junior life scientists across three years of programming.

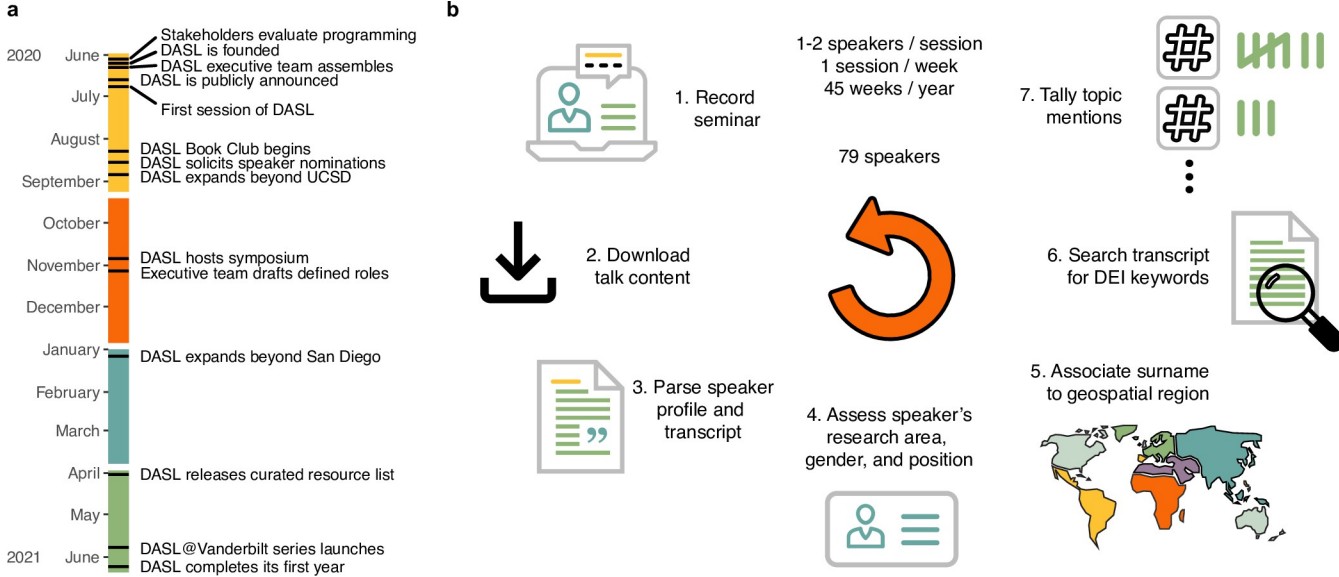

**Fig 1. Reviewing the first year of DASL seminars.** a) A timeline of DASL milestones separated by quarter (yellow = summer, red = fall, blue = winter, green = spring). DASL was founded in June 2020 and completed its recent spring series in June 2021. b) Outline of the approach taken to synthesize insight from 79 weekly DASL seminar series speakers. World map colors specify geospatial regions (green = Europe, light green = Anglophone, yellow = Latin America, Spain, Portugal & the Philippines, red = Africa, purple = Middle East & North Africa, blue = Asia).

Who claims the opportunity afforded by a new diversity-themed seminar series like DASL, and what are speakers' priorities for DEI in STEM?

Answering such questions can inform further efforts to build platforms for DEI in STEM and translate a transient awareness of DEI concerns into enduring policy changes. An understanding of what candidate speakers value and which facets of diversity may be absent greatly facilitates speaker recruitment. After hosting speakers, documentation of recurrent concerns and recommended remedies can point to both areas in need of restructuring or programs meriting expansion.

Yet, seminars typically consist of unstructured speech. Coding speaker profiles and speech into quantitative data entails tremendous discretion and diminishes speakers' qualitative experiences, but basing decisions on qualitative impressions makes effective prioritization of goals difficult and risks inaction on comparatively important but less salient concerns.

Reproducible statistical analysis for diversity requires special care. Only a few identities, such as nationality, geography, and gender, are commonly public information. Self-identification promises low rates of misclassification but still poses logistical challenges: what, if any, prefilled options to offer, and how to handle missing data, spelling errors, misunderstandings, and possible identifiability of participants if results are published. Programmatic and semi-automated methods that make inferences based on names or documents have the potential to curtail response bias, protect participant privacy (e.g., no exposure of phone number, mailing address, or email address), and rescue data lacking self-identification. Indeed, participant name is sufficient for scalable and reproducible (if uneven) inference of race, nationality, gender, and geographic associations [8–12]. However, some identities, such as class, rural or agricultural background, first generation status, and sexual and gender minorities are almost wholly inaccessible without self-identification.

Here, we summarize how trainee life scientists conceive of DEI in STEM (Fig 1B). We aggregated gender identity from speaker presentations, algorithmically inferred geography from speaker surnames, and topic mentions gleaned from seminar transcripts to produce highly powered reproducible datasets not limited in scope by preconceived survey questions or speaker response rate in post hoc self-reported information. We find large differences in representation compared to a traditional seminar series and identify differences in topic noteworthiness that vary by speaker background and priorities. We examine self-reported reasons junior life scientists give to participate in a new diversity-themed seminar series as well as the topics most often discussed as assessed from seminar recordings. We review resources that DASL speakers cited as personally helpful and characterize how different minority populations may be best served by different types of DEI programming. Our work provides a template for future computational audits of diversity in STEM that provide actionable insight for speaker recruitment and priorities for DEI policy changes.

## Methods

### Speaker recruitment

Summer quarter DASL speakers and all faculty speakers were recruited directly by DASL organizers representing the graduate programs in Biological Sciences, Biomedical Sciences, Bioinformatics and Systems Biology, Bioengineering, and Neurosciences, plus postdoctoral scholars appointed in UC San Diego's School of Medicine and School of Biological Sciences. In subsequent quarters, nominations were accepted through a Google Form posted on the DASL website and shared on the DASL email list. Nominated trainee scientists were contacted and scheduled if the DASL platform appealed to them.

## Speaker profile analysis

Terms in talk titles were counted by parsing titles posted on the DASL website using the R packages tidytext and SnowballC. Text was tokenized by tidytext::unnest_tokens and stop words removed using the tidytext's stop_words dataset. Words were stemmed using SnowballC::wordstem, and the number of titles containing each stemmed word were counted. The frequency of retained stems that occurred in at least 3 titles were converted back into representative words and visualized using the wordcloud package.

Representation of women was computed from UCSD diversity dashboards. Graduate students in the Health Sciences and Biological Sciences who chose "Woman" as their gender were counted. For postdocs, academic personnel with the appointment title "Postdoctoral scholar/ fellow" in the Health Sciences and Biological Sciences were counted. For faculty, the "ladder-rank professor" appointment title was selected. Data was taken from fall 2019, the most recent data available. Probability of success in null binomial testing was weighted by the fraction of women at each career stage (50.7% women overall).

## Geospatial name analysis

Speaker surnames were noted from the DASL website and Fragile Nucleosome website over the same time period. Surnames were associated to geographic regions using Forebears.io, the "largest geospatial names database", which purportedly aggregates records from over 27 million surnames and 4 billion individual records from 236 countries or jurisdictions [13]. Jurisdictions in Latin America were pooled into one region. Jurisdictions in Britain, the United States, Canada, Australia, New Zealand and the Caribbean were pooled into an Anglophone region because of their shared British surnames [14, 15]. South Asian jurisdictions were pooled into a South Asian region. For each surname, the region with the highest incidence and frequency were noted. Names from speakers with two surnames were weighted accordingly. For surnames that matched multiple regions, the most frequent region only applied when the incidence was above 500.

Surnames were considered validated under one of the following conditions: 1) if the most prevalent region was also the most frequent, 2) if the most prevalent region was at least tenfold greater than the most frequent region (most prevalent region chosen), or 3) if the most prevalent region was the Anglophone region and less than tenfold greater than the most frequent region (most frequent region chosen). If these conditions were not met, or there was no match to the database, the surname was deemed unidentified. Surnames assigned to the Anglophone region were further annotated with race using the predictrace::predict_race command from the predictrace R package. Surnames assigned high confidence for matching White US Census takers were labeled "(White)". Surnames that were not high confidence for corresponding to White US Census takers were labeled "(Nonwhite)". Note that these methods are not sensitive for Native American and Alaskan Native or multiracial identities reported to the US Census [16].

We compared our geospatial name inferences to output from a recently published long short-term memory model called Wiki2019-LSTM [8]. The Wiki2019-LSTM model was downloaded from its GitHub repository (https://github.com/greenelab/wiki-nationality-estimate). The 07.test-ismb-data.py script was modified to issue predictions on DASL speaker surnames. For aggregate counts, fractional assignments were summed across all speakers. For visualization of the confusion matrix, surnames that had no region above 50% probability were labeled as "Unidentified". For the sake of comparison with Forebears lookup, the Wiki2019-LSTM's Celtic/English category was deemed synonymous with our Anglophone category, and the Wiki2019-LSTM's Israel category was pooled with our Middle East and North Africa category.

## Transcript topic analysis

Topics and keywords for quantitative content analysis were developed over multiple biweekly DASL organizer meetings. Over ten trainee life scientists from disparate graduate programs and departments participated and brainstormed topics. Draft results were shown and topics were added or subdivided per organizer feedback until the organizers were satisfied that the topics reflected the breadth of content covered in DASL seminars. For example, organizers decided to add a failure topic (keywords: struggle, challenge, fail, overwhelmed), and organized decided to divide socioeconomic divided into class (keywords: poor, poverty, income) and finances (keywords: pay, money, afford) (S1 File).

Publicly viewable DASL and SQUAD talks were downloaded and then uploaded to You-Tube, which automatically generated English language captions. Downloaded captions served as talk transcripts. We observed higher fidelity using YouTube than competing automated services such as Zoom, consistent with reports that YouTube produces nearly half as many errors in its audio transcripts: 5 words out of 100 for YouTube versus 8 for Zoom [17]. Transcripts were reviewed and keywords tallied in R. Exact keyword expressions and counts are available in the S1 File. Mentions of "white matter" and other brain science terms were manually removed from counts of "White" mentions. Column order was determined by running the R command hclust on Pearson correlation distance for a matrix of the log10 of counts plus 1 which was then scaled with the scale command for each keyword across 54 talks. The number of mentions of each keyword were truncated at 3 or more for data visualization purposes.

Significantly correlated topics or survey answers were determined using the R command cor(m ="s") on the matrix of total mentions by speaker and keyword and removing duplicate comparisons; p-values were computed from an empirical distribution using the ecdf command on permuted data. Self-reported themes and interests were scored according to their rank (rank 1 = 3, rank 2 = 2, rank 3 = 1). Themes and interests were not tested against each other because speakers are limited to three each such that choosing one theme or interest precludes selection of another. Differential mentions by speaker status and seminar series were determined by using the R command binom.test on the sum of truncated mentions (DASL speakers with more than 3 mentions were replaced by 3 and SQUAD speakers with more than 6 replaced by 6 to mitigate the impact of outliers) split by gender, degree status or seminar series. The false discovery rate was set using the p.adjust command. Binomial confidence intervals were obtained using the binconf command from the Hmisc R package.

## Google scholar search

The Google Scholar search queries "allintitle: diversity (race OR racial OR ethnic OR color)", "allintitle: diversity (mother OR mothers OR father OR fathers OR son OR sons OR family OR families)". and "allintitle: diversity (mentor OR mentors OR mentorship OR mentoring)" were used to estimate the relative focus of diversity literature on race, family and mentoring.

## Results

In summer 2021, DASL concluded its first year of programming (Fig 1A). What once started as a community at UC San Diego grew to include scientists affiliated with universities and research institutes nationwide. Sustaining open and impactful conversation amongst graduate students, postdoctoral scholars, and faculty who had never met in person entailed immense planning and effort (S1A Fig), but over its first year, DASL continuously grew its subscriber base (S1B Fig), accrued over 4,000 unique website visitors, and achieved roughly 3,500 hours of engagement from attendees (S1C Fig).

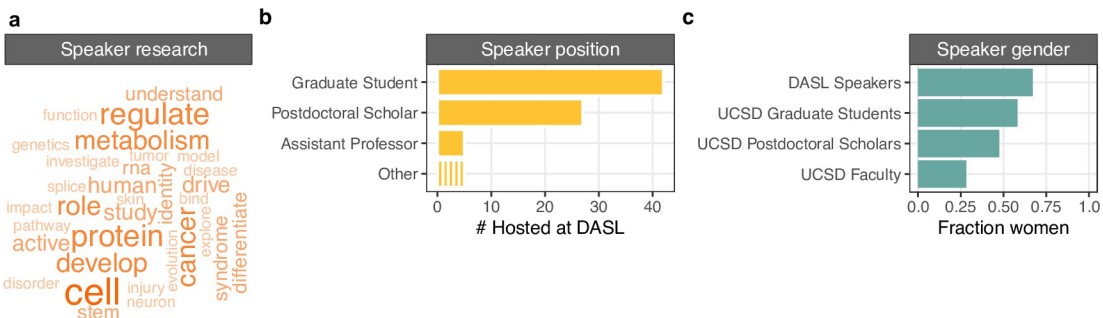

**Fig 2. Analysis of DASL speaker profiles.** a) Word cloud of the most common terms used in DASL research talk titles. The size and shade of the word reflects the frequency. b) Breakdown of the positions held by DASL's first 79 speakers. The 'Other' category includes Research scientist, Research Assistant Professor, Professor, Assistant Curator, and Administrator. c) Representation of women among DASL speakers compared to UCSD life science graduate students, postdoctoral scholars and faculty.

Over the course of a year, DASL hosted 79 speakers. First, we summarized the research focus of DASL speakers by parsing keywords in talk titles from speaker profiles. "Cell" was the most used term, at nine mentions, followed by "protein" and "regulate" at seven mentions, and "cancer", "develop", and "metabolism" at six. Other top words typically related to human cellular and molecular biology (Fig 2A). Next, we examined information about the speakers themselves. The majority of speakers were graduate students (53%), followed by postdoctoral scholars (34%), and assistant professors (6%) (Fig 2B). Overall, a greater percentage of speakers were women (67%) relative to UCSD life scientists broadly (58% of graduate students, 48% of postdocs, and 28% of tenure-track faculty; Fig 2C). By comparison, the UCSD Cellular & Molecular Medicine departmental virtual seminar series featured 42% women professors in 2021 (n = 31). Overall, women were more likely to participate in DASL than men (odds ratio 1.9, p = 5e−3, one-tailed binomial test).

## Speaker surnames map a geography of DASL speaker diversity

To further characterize the diversity of speakers, we determined the country or region in the world where each speaker's surname was most prevalent and abundant as reported by the Forebears database (see Methods). For most names, prevalence and abundance pointed to the same region, enabling imperfect but scalable inference of where relatives of speakers currently reside. (Fig 3A). To reflect historical global migrations and corresponding name usage, names linked to Britain, the United States, Canada, Australia, New Zealand and the Caribbean were grouped in a separate Anglophone category—surnames that were most abundant in Anglophone countries but more prevalent in a non-Anglophone region were annotated in accordance with the non-Anglophone region.

Overall, Latin American (30 names) countries were the top associated regions for DASL speakers' surnames. Next most common were Anglophone (20 names) and European countries (13 names) (Fig 3A). We further annotated anglophone-associated surnames as "White" or "Nonwhite" using US Census data linking common US surnames to race [10]. Using this approach, 82% of Anglophone speaker surnames were labeled "Nonwhite" in DASL.

We performed the same analysis for 63 speakers hosted by the Fragile Nucleosome forum, a concurrent international remote lecture series that has promoted community building (S2 Fig) [7]. 30 Fragile Nucleosome speaker names were associated to European countries and 14 to Anglophone countries. Most Anglophone surnames (64%, n = 14) were labeled "White." Overall, 30% of Fragile Nucleosome speaker surnames were associated to regions outside Europe and Anglophone regions compared to 58% for DASL speaker surnames.

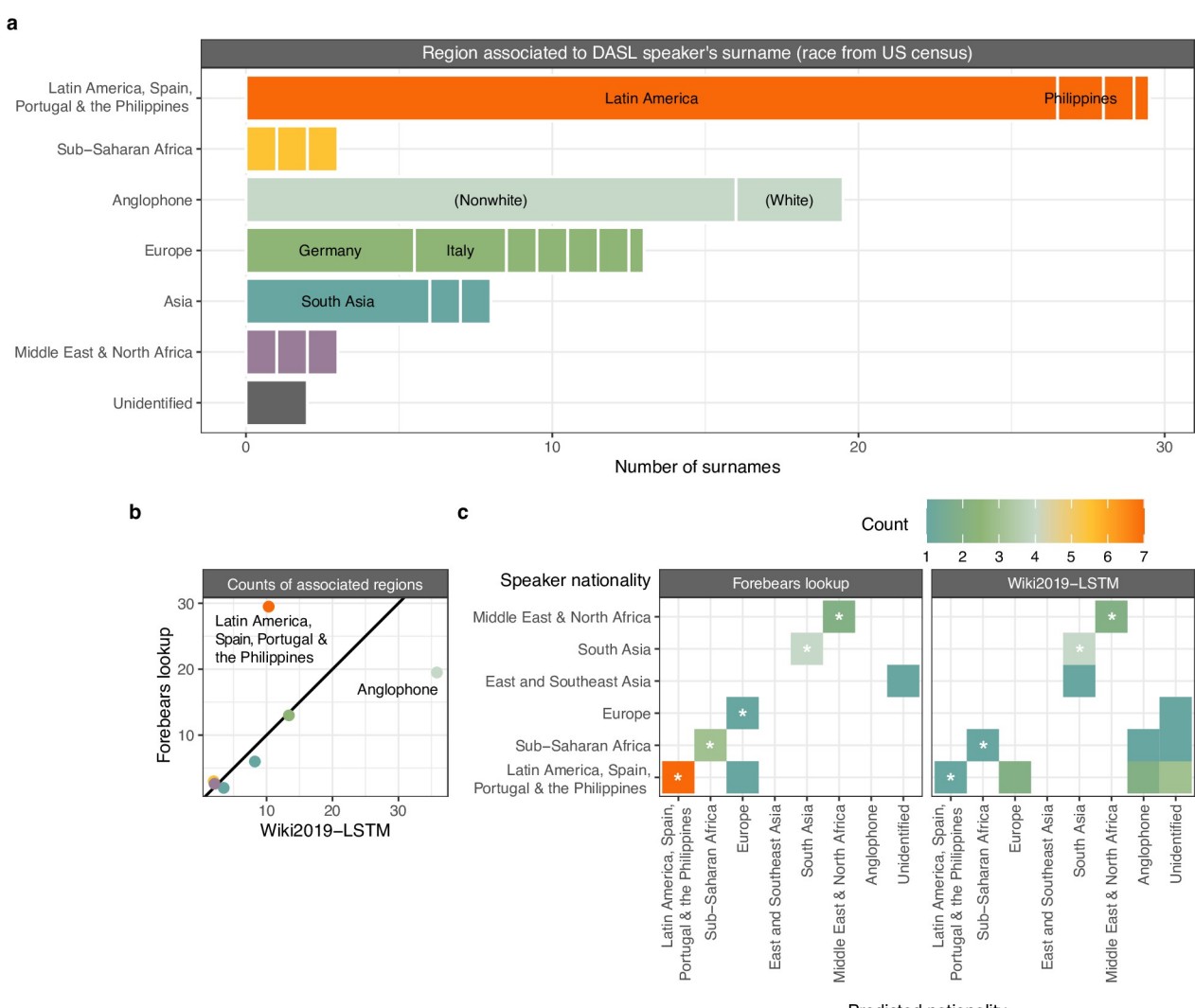

**Fig 3. Association of DASL speaker names to geospatial groups.** a) Counts of most associated regions for DASL speakers' surnames using the geospatial name database Forebears. 'Anglophone' refers to names associated to Britain, the United States, Canada, Australia, New Zealand or a country in the Caribbean. b) Aggregate counts for associated regions using Wiki2019-LSTM (x-axis) and the Forebears database (y-axis) across all DASL speakers. c) Confusion matrices for international DASL speaker names (19 surnames from 14 speakers) demonstrating accuracy of predictions for speaker nationality. Asterisks denote correct identifications. Names for which Wiki2019-STLM achieved less than 50% probability for a region were labeled as "Unidentified".

We next compared lookup of surname-associated regions in Forebears to imputation using a recently published long short-term memory recurrent neural network trained on names and nationalities scraped from Wikipedia ("Wiki2019-LSTM") [8]. For most geographic regions, predictions agreed well in aggregate, but Anglophone names and names associated to Latin America, Spain, Portugal, and the Philippines were discrepant (Fig 3B).

We re-examined predictions using publicly declared nationalities from select speakers as well as racial identities of speakers posted on the DASL website. For the 14 speakers who publicly identified their nationality, wefound a 71% success rate for lookup in Forebears versus 36% for LSTM predictions. LSTM predictions for Latin American speakers' names were often linked to Anglophone regions (Fig 3C), suggesting that overrepresentation of contributors from Anglophone countries in Wikipedia may cause bias against Latin American names in the

learned LSTM. On the DASL website, two speakers preferred not to disclose their race, but 29 out of 79 speakers reported their racial identity on the DASL website (S3 Fig). Of those, 1 was unidentified and 26 had a matching associated region or census-based inference (93% accuracy). Of note, no speakers associated to the Middle East or North Africa reported data to the DASL website. Surname association by direct lookup with racial disambiguation by census data excelled in annotating all speakers' backgrounds.

## DASL speakers emphasize the importance of social and interpersonal factors in STEM

DASL speakers covered a broad range of topics: childhood familiarity with a career in STEM, the burden of fees in graduate admissions, immigrant identity, coming and being out as queer or trans in academia, navigating parenthood in STEM, mental health challenges in academia, health disparities for racial minorities, advocacy for people with disabilities, the complexity of multi-racial identity, cultural expectations clashing with career pressure, anti-science attitudes in rural America, and both subtle and overt racism against Black people and people of Middle East and North Africa descent.

To summarize DASL speakers and talk content, we analyzed the identities and motives of 36 DASL speakers posted on the DASL website as well as the themes each speaker chose to describe their talk (S4 Fig). First generation status was claimed by nearly half of the speakers (17/36, S4A Fig). Low socioeconomic status, immigrant identity, LGBTQ+ identity, and rural and agriculture background were each claimed by approximately a quarter of the speakers. An overwhelming majority of speakers (33/36) wanted to contribute to discourse on DEI, and nearly two-thirds (23/36) selected it as the top reason. In descending order, mentorship, women in science, and the importance of education were the most commonly chosen themes. Especially salient themes included women in science, sexual and gender minorities, and rural and agricultural background, each of which were the top theme for half of speakers who chose it.

We also tested for co-occurrence across 23 identities, motives, and themes listed by DASL speakers and called 39 correlated or anticorrelated responses (permutation of spearman correlation coefficient, 10% FDR, S4B Fig). Some pairing of identities and themes were wholly predictable: woman identity to the theme of women in science, LGBTQ+ identity to the theme of sexual and gender minorities, rural identity to the theme of rural and agriculture background, and low socioeconomic status to the theme of financial barriers. Some co-occurring identities pointed to racial disparities: low socioeconomic status with Hispanic/Latino, American Indian/Alaska Native, and multiracial identities, disability with American Indian/Alaska Native race, and mental health for white race. In some cases, co-occurrence suggested diverging attention across identities: Women speakers less often discussed sexual and gender minorities or rural and agricultural background, and Black speakers expressed more interest in DASL for the opportunity to present their research.

We next analyzed themes in DASL talks in more detail by counting mentions of keywords aggregated by topic for each trainee DASL talk with an accessible recording (N = 54) (Fig 4A). Topics related to social factors were most broadly discussed. "Education" keywords were most frequently mentioned (46/54 talks), followed by "Family" keywords (39/54) and "Mentoring" keywords (37/54). Keywords for the concepts "Collaboration" and "Success" were also mentioned by a majority of speakers, but typically received few mentions per speaker. Other topics were mentioned many times by a small number of speakers: "Sexual and gender minorities", "Mental health", and "Finances". Mentions of "Race/ethnicity" and "Friends" were intermediate in both regards: speakers broke roughly evenly between zero mentions, one or two mentions, or many mentions. Thus, as with the themes self reported by DASL speakers, the

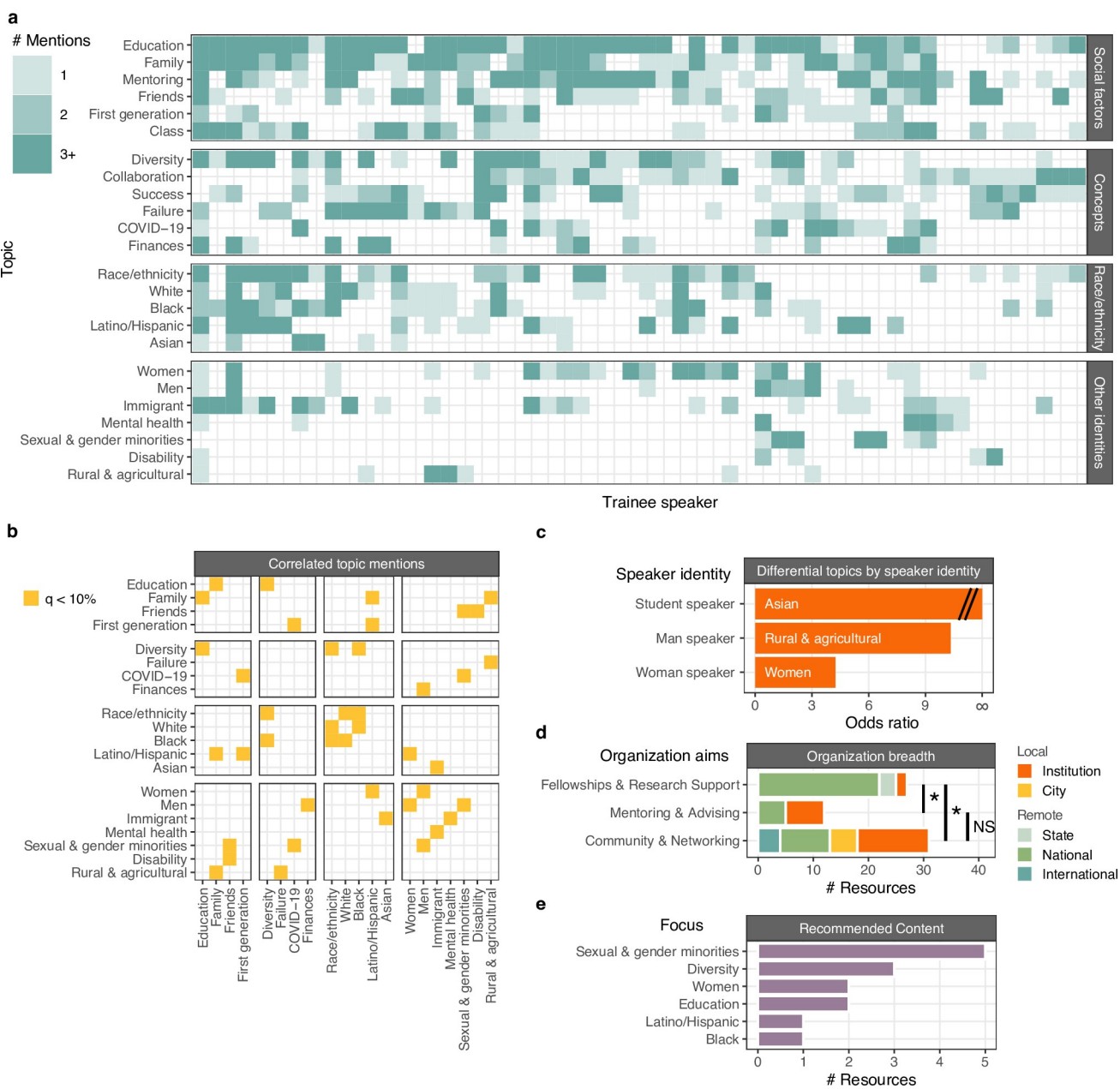

**Fig 4. Topics relevant to diversity, equity, and inclusion mentioned by trainee DASL speakers.** a) Truncated counts (more than 3 mentions were placed in "3+") of mentions of topics across speakers (x-axis) organized by topic (y-axis) and broader type of concern (social, race/ethnicity, or other identities). b) Significantly correlated keyword usage for terms in a). c) Differential keyword use stratified by speaker identity. d) Helpful organizations cited by DASL speakers, annotated by organization aims and scope. e) Count of DASL speakers' recommended content stratified by focus.

noteworthiness of topics assessed from talk transcripts varied considerably both in breadth (fraction of speakers mentioning a topic) and salience (the typical number of mentions per speaker).

## Topics discussed vary by speaker identity and intersect with one another

21 pairs of topics exhibited correlated keyword usage (permutation of spearman correlation coefficient, 10% FDR) across 54 talks (Fig 4B). The most significantly correlated topics were

"White" and "Race/ethnicity" (rho = 0.467, p = 4e−4), closely followed by "Diversity" and "Education" (rho = 0.465, p = 8e−4). Whereas "Friends" correlated with "Sexual & gender minorities" and "Disability", "Family" correlated with "Latino/Hispanic" and "Rural & agricultural". Indeed, 100% of talks mentioning "Latino/Hispanic" keywords also mentioned family, strongly linking discussion of Latino and Hispanic identity to family values. Conversely, disability and sexual and gender minority identities were linked to noteworthiness of friendship.

Other correlated themes supported nuanced interpretation of the data. Mentions of "Immigrant" correlated with "Asian" but not other racial groups, and "Latino/Hispanic" with "Women" but not men, suggesting the existence of special intersectional identities. Similarly, mentions of "Sexual & gender minorities" and "Finances" were correlated with "Men" but not "Women" keywords. Mentions of "First generation" correlated with "COVID-19", which suggests that disruptions from the pandemic to higher education might be particularly salient for those who are first in their family to attend. These relationships were not evident from the limited data speakers self-reported on the DASL website.

In a few cases, the likelihood of a topic depended on the degree status or gender of the speaker (Fig 4C). Men speakers were 10 times more likely to mention "Rural & agricultural" keywords than women speakers (Binomial test, p = 3e−4), women speakers were 4 times more likely to mention "Women" keywords than men speakers (Binomial test, p = 4e−4), and graduate student speakers were more likely to mention "Asian" keywords than postdoctoral scholars (Binomial test, p = 2e−3, 10% FDR). We expect that smaller but still significant differences in topic mentions across speaker identities could be detected with a larger pool of speakers.

## Speaker's resources suggest social bonds are vulnerable to disruption by social distancing

In addition to keyword analysis, we curated specific resources DASL speakers cited as helpful to their scientific careers. The 90 entries include entities offering fellowship and research support, community and networking groups, mentoring and advising centers, professional societies, and recommended content [18]. Cited mentoring and advising entries (Fisher exact test, p = 6e−5) and community and networking entries (Fisher exact test, p = 1e−3) were 16 times more likely to operate at local levels compared to fellowship and research support opportunities (Fisher exact test, p > 0.05) (Fig 4D). While funding sources may be fungible and robust to social distancing, we infer that community-building and mentoring opportunities that trainee scientists relied on were greatly diminished by pandemic precautions.

For the 14 recommended websites and articles ("Recommended content"), we labeled each with the most relevant topics describing their content (Fig 4E). The plurality (5/14) commented on sexual and gender minorities, followed by diversity broadly (3/14) and women and education (each with 2/14).

## Validating topic mention counts with self-reported data

Topics appeared to vary in breadth and salience across DASL talks (Fig 5A). To confirm that the correlations we reported were not an artifact of common words or misassignment of topics from keywords, we performed a similar transcript-based analysis for the 18 talks with corresponding self-identified data from the speaker on the DASL website using a relaxed 20% FDR threshold. We detected 16 topics that varied across speaker race, other identity, or interest or talk theme and found strong evidence that talk content directly reflects speaker identity: Hispanic/Latino and Asian speakers more often mentioned their respective races, immigrant, LGBTQ+, and rural background speakers more often mentioned their respective identities, and low socioeconomic status identifying speakers more often mentioned financial concepts (Fig 5B). We confirmed

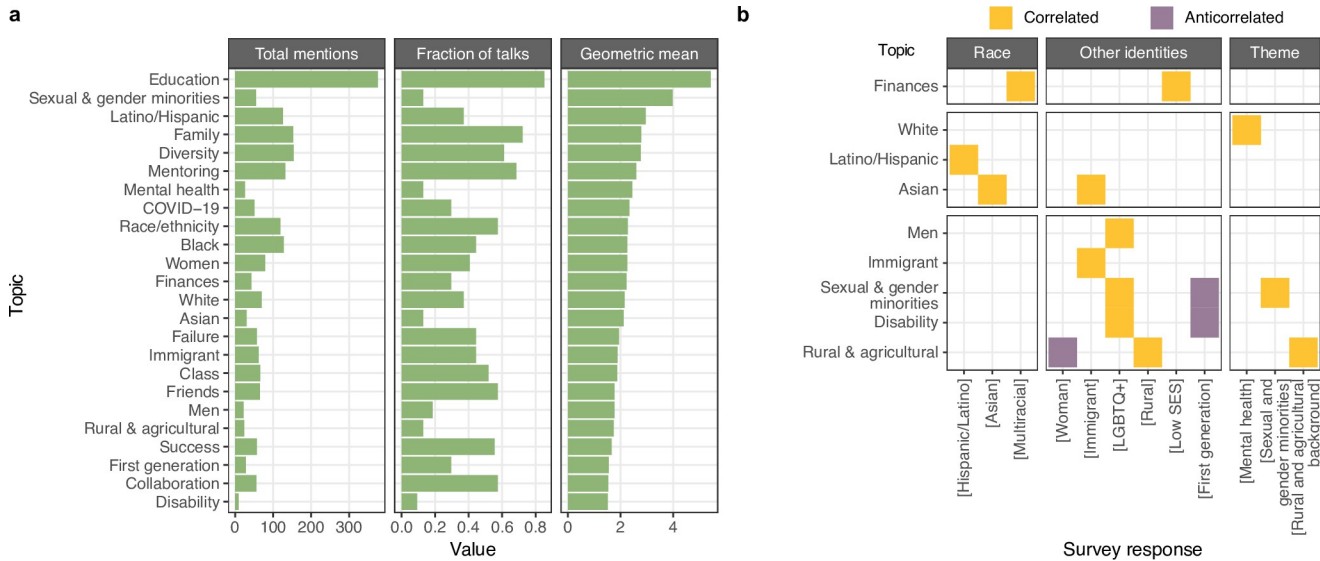

**Fig 5. Quantifying and contextualizing topic noteworthiness at DASL.** a) Count of the total mentions per topic (left), topic breadth, or the fraction of talks with at least one mention (middle), and topic salience, or the geometric mean of the number of mentions per talk with at least one mention (right). b) Significant co-occurrence of topics derived from talk transcripts with self-reported identities and talk themes from DASL speakers.

intersectional concerns for immigrant speakers discussing Asian identity, and LGBTQ+ speakers discussing male identity, giving credence to the transcript-only results reported above.

## DEI speakers' focus on social and interpersonal factors persists across seminar series

We applied the same keyword analysis approach to another UC San Diego DEI seminar series: Scientific Queers United in Academic Discourse (SQUAD). In contrast to DASL, SQUAD features approximately 40-minute talks exclusively from LBGTQ+ scientists based in San Diego and does not seek speakers from a specific career stage or discipline. SQUAD speakers do address both their research and their personal experiences, as in DASL seminars.

Unsurprisingly, SQUAD speakers most often mentioned "Sexual & gender minorities" keywords—mentions were 8.5 times more likely than in DASL seminars (Fig 6). Otherwise, SQUAD speakers largely mentioned the same social and interpersonal factors highlighted by DASL speakers. The next most frequent topics at SQUAD after "Sexual & gender minorities" were, in order, "Education", "Friends", "Mentoring", and "Family" (Fig 6).

We next assessed more subtle differences in keyword usage between DASL and SQUAD. SQUAD speakers were 4 times more likely to mention "Asian" keywords (Binomial test p = 5e−4), 2.1 times more likely to mention "Friends" keywords (Binomial test p = 1e−3), and 4.3 times less likely to mention "Latino/Hispanic" keywords (Binomial test p = 5e−3), after applying a 10% FDR (Fig 6). The greater use of "Friends" keywords in SQUAD seminars echoes our earlier observation of correlated usage of "Sexual & gender minorities" and "Friends" keywords. Notably, SQUAD speakers did not uniformly increase mentions of social factors: family keyword use was less frequent, if anything (p > 0.05). Transcripts of SQUAD seminars are thus further evidence of greater relative attention paid to friendship among sexual and gender minorities.

## Discussion

In this work, we examined 79 speaker profiles and 54 talk transcripts from the Diversity and Science Lecture series. We characterized who spoke at DASL and the topics they discussed

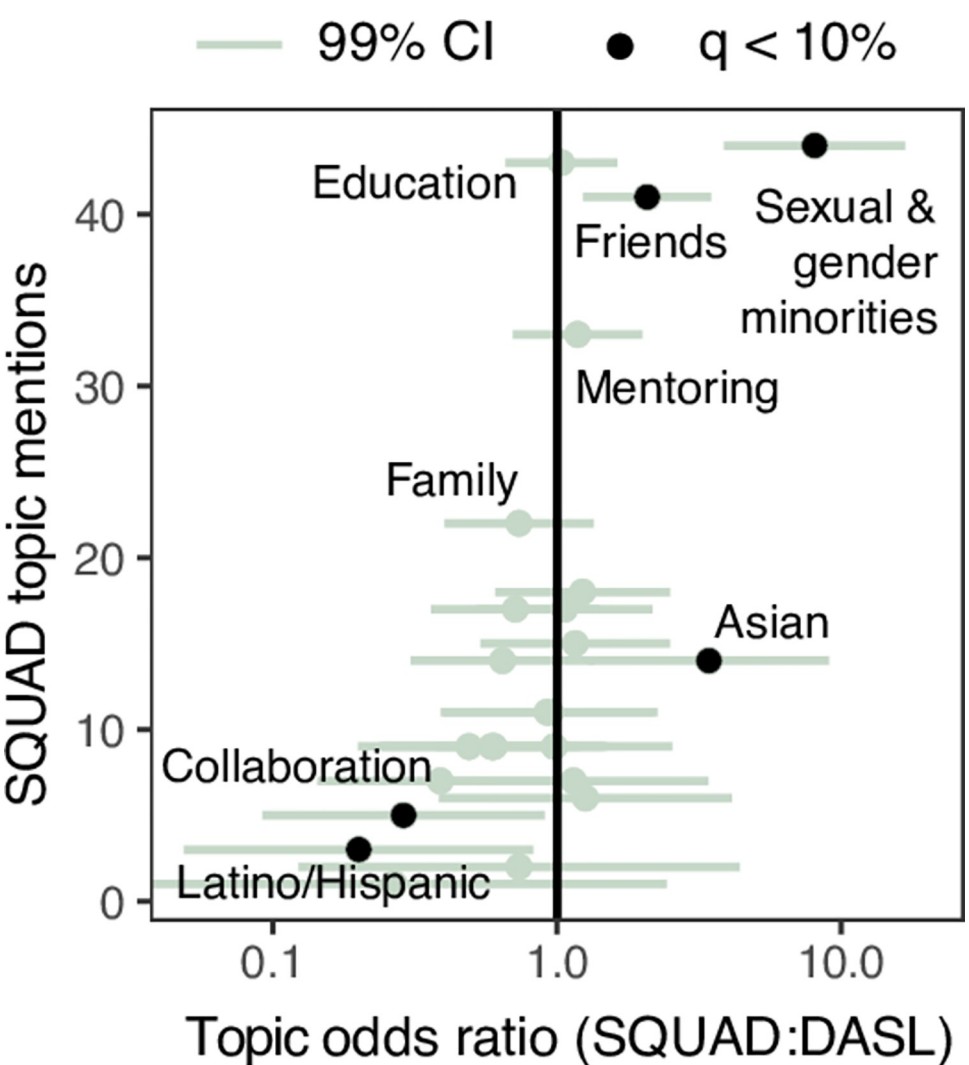

**Fig 6. Comparison of DEI topic mentions between DASL and SQUAD.** The odds ratio for each topic mentioned by a SQUAD speaker versus DASL and its 99% confidence interval are shown.

using a quantitative content analysis framework. We cross-referenced the gender balance of DASL speakers against that of UCSD life science trainees and found greater participation from women scientists. We also exploited a geospatial analysis of speaker surnames that showed strong engagement with Latino and Hispanic scientists that far surpassed that seen in Fragile Nucleosome, a non-DEI life science seminar series (S2 Fig). Our tallies of keyword mentions in talk transcripts indicated that social factors like family, education, and mentorship are highly salient and broadly noteworthy (Fig 5A). Sexual and gender minority was noteworthy to fewer life science trainees but still highly salient. Our review of content recommended by speakers addressed offered more evidence for sexual and gender minority identity to select trainees (Fig 4E).

Our audit of speaker diversity yielded actionable insight into the operation of DASL. Men and Asian DASL speakers—especially Asian men speakers—were poorly represented. Only graduate students discussed Asian identity amongst recorded talks, suggesting the need for greater visibility of Asian postdoctoral scholars. Comparison to a SQUAD, a companion DEI

series on LGBTQ+ scientists, further emphasized the dearth of Asian speakers at DASL (Fig 6). Men speakers disproportionately commented on finances, sexual and gender minorities, and rural and agricultural background, suggesting that a greater focus on these aspects of diversity may be a way to engage more men. Our semi-automated approach based on the Forebears geospatial names database eliminated errors and irreproducibility due to subjective classification of speaker backgrounds and simplified comparisons across platforms. We also note that the Forebears API can process thousands of surnames near instantaneously with reduced data entry burden. In response to these results, DASL organizers intentionally recruited Asian and men speakers for its second year of seminars.

We further learned about effective speaker recruitment for diversity-themed seminar series. We found that contributing to discourse on diversity, equity, and inclusion was highly motivating for junior life scientists (S4A Fig). Furthermore, Speakers reported unanimous approval for dry runs that allowed them to receive feedback on their presentations before formal sessions. Based on DASL speakers' self reported interest, we expect that similar programming focused on mentorship would achieve high engagement. Ultimately, disparities for women [19–23], racial minorities [8, 24–27], and sexual and gender minorities [28–31] working in scientific disciplines have been well documented, and a seminar series offers a path for continued visibility.

The most noteworthy topics in DASL talks betrayed our expectations. Junior life scientists' reflections on the contribution of interpersonal relationships to DEI in STEM were timely amidst the unprecedented social distancing measures implemented early in the COVID-19 pandemic. Still, it is notable that discussion of social factors including education and mentorship surpassed that of racial identity in a diversity-themed seminar series, as assessed by analyzing talk transcripts (Fig 5A) and the speakers' self-reported intent (S4A Fig). Academic literature on diversity broadly, beyond the purview of trainee life scientists, addresses race and ethnicity (9750 articles in a Google Scholar search) much more often than family (4040 articles) or mentoring (445 articles). Furthermore, the noteworthiness of social factors like education, family, and mentoring stood out not only because nearly all speakers addressed them, but because each speaker commented on them repeatedly (Fig 5A). Interestingly, speakers pointed primarily to national sources of funding as personally helpful, suggesting a disconnect between the fungible nature of funding and the interpersonal support desired by trainee life scientists (Fig 4D). We believe that trainee life scientists highly esteem a sense of camaraderie and that access to interpersonal support is essential for achieving DEI goals at academic institutions even when funding levels are ample.

Our correlational analysis suggests special care be taken regarding intersectional identities and common needs within historically excluded groups. Particularly in light of recent anti-Asian sentiment in North America, Asian immigrant identity comes with unique burdens that cannot be recapitulated from separate Asian and immigrant identities. Similarly, Latina women face specific stereotypes that do not burden women or Latino/Hispanic people broadly.

The social ties highlighted by speakers clearly varied by the identities discussed, with concomitant policy implications. Friends and family were each mentioned by a majority of speakers, but friendship garnered more commentary in talks mentioning sexual and gender minorities and disability, and family garnered more commentary in talks mentioning Latino/Hispanic and rural background. We hypothesize that sexual and gender minorities and people with disabilities on average pay greater attention to friendship in order to connect with those with a shared identity. By comparison, Latino/Hispanic individuals and those from rural backgrounds on average pay greater attention to family, who likely share in those identities. The visibility granted by these concerns at DASL have direct policy implications: LGBTQ+ and disabled identifying groups may prioritize access to community space and funding for social

events whereas Latino/Hispanic identifying groups may prioritize access to child care subsidies and family leave. An overly narrow approach to academic programming may leave certain minority groups dissatisfied given their backgrounds.

The methods we present do not solve all challenges regarding data on diversity in STEM. Aggregating identities with divergent lived experiences into broad categories ("Asian" or "Sexual and gender minorities") can mask important disparities for subgroups (Southeast Asian or transgender identity), but low counts for more rarely adopted or hard-to-define identities preclude statistical inference. Indeed, multi-racial identity could only be studied using speakers' self-reported information. Attention paid to each topic is likely influenced by the seminar organizers involved and the employed recruitment strategy. Finally, balancing topics of high salience to a small subset (e.g., sexual and gender minorities) against topics of modest salience to a majority (e.g., the role of collaboration in research) is challenging but necessary to realize an inclusive environment in STEM.

In conclusion, we believe that harnessing speaker profiles and transcripts from scientific seminars for content analysis can inform engagement of speakers and organizers for future initiatives and guide funding and programming decisions for advancing diversity, equity, and inclusion in STEM.

## Supporting information

**S1 Fig. Engagement in the first year of DASL.** a) Outline of running a remote lecture series. We provide guides detailing how to approach each underlined component on our website. b) Total subscribers to the DASL mailing list over time. c) The number of attendees per DASL session over the first year of programming.
(PDF)

**S2 Fig. Fragile Nucleosome speaker surname associations.** Speaker name associations from the Fragile Nucleosome seminar series. Counts are tallied as in Fig 3A.
(PDF)

**S3 Fig. Self-reported race of DASL speakers.** Counts of racial identities separated by speakers who identify as a single race or multiracial. Labels were taken directly from the Speakers page of the DASL website.
(PDF)

**S4 Fig. Other self-reported data from DASL speakers.** a) Counts of speakers who selected each identity, interest, and talk theme. The top ranking talk theme selected by the speaker is indicated in a darker shade of green. b) Correlated and anticorrelated identities and talk themes for DASL speakers who posted data to the DASL website.
(PDF)

**S1 File. Information on authors and all data mined from seminars.** DASL Alliance author list, surname associations for DASL and Fragile Nucleosome, search queries for DEI topics, topic mentions in talk transcripts for DASL and SQUAD seminars, annotated resources cited by DASL speakers, and DASL speaker survey responses and biographical data mined from the DASL website.
(XLSX)

## Acknowledgments

We thank the HHMI Gilliam Fellows, the ThermoFisher Queer Working Group, the Huntsman Cancer Institute, the UCSD Biology Student Social Fund, UCSD Graduate Division,

UCSD Biomedical Sciences Graduate Program, the UCSD Biology Diversity Committee, the UCSD Office for Equity, Diversity, and Inclusion, the UCSD Latinx/Chicanx Academic Excellence Initiative, the UCSD Black Academic Excellence Initiative, and the UCSD School of Medicine for supporting the DASL 2020 symposium. Icons for the DASL schematics were downloaded from The Noun Project (nounproject.com). Lightbulb by Maxim Kulikov, Rules by Adrien Coquet, Webinar by ProSymbols, Whisper by ProSymbols, Toolbox by WEBTECHOPS LLP, Document by Jamison Wieser, downloads by Gregor Cresnar, Count by Deylotus Creative Design, redo by Creative Stall, Magnifying Glass by Chatchai Pripimuk, hastag by Tomi Triyana, identity by Saurus Icon. We thank the oSTEM UCSD graduate chapter leadership for hosting SQUAD and making seminars available on their website. DASL Alliance authors: Andres Nevarez (UCSD), Homa Ghalei (Emory), Tori Placentra (Emory), Grace Xiao (UCLA), Jason Shepherd (University of Utah), Terry Gaasterland (UCSD), Emma Farley (UCSD), Stacey Brydges (UCSD), Michelle Ragsac (UCSD), Luca Caputo (Sanford Burnham Prebys), Stephen Sakuma (Sanford Burnham Prebys), Angela Nicholson-Shaw (UCSD), Lydia Deboussi (Salk Institute), Jenny He (UCSD), Shobha Vasudevan (Massachusetts General Hospital), Catherine M. Gavile (University of Utah), Asa Gustafsson (UCSD), Cosimo Commisso (Sanford Burnham Prebys), Mary Catanese (Massachusetts General Hospital), Nuttida Rungratsameetaweemana (Salk Institute), Christian Cazares (UCSD), Kellie Williford (Vanderbilt University), Ethan Lippmann (Vanderbilt University), Cindy Barba (University of Utah), Evan Boyle (UCSD).

## Author Contributions

**Conceptualization:** Evan A. Boyle, Gabriela Goldberg, Jillybeth Burgado, Hannah A. Grunwald, Aleena K. S. Arakaki, Gene W. Yeo.

**Data curation:** Evan A. Boyle, Gabriela Goldberg, Jonathan C. Schmok, Jillybeth Burgado, Fabiana Izidro Layng, Hannah A. Grunwald, Kylie M. Balotin, Michael S. Cuoco, Keng-Chi Chang, Aleena K. S. Arakaki, Noorsher Ahmed, Ximena Garcia Arceo, Pratibha Jagannatha, Jonathan Pekar, Mallika Iyer.

**Formal analysis:** Evan A. Boyle, Keng-Chi Chang.

**Funding acquisition:** Evan A. Boyle, Gene W. Yeo.

**Investigation:** Evan A. Boyle, Gabriela Goldberg, Jonathan C. Schmok, Jillybeth Burgado, Fabiana Izidro Layng, Hannah A. Grunwald, Kylie M. Balotin, Michael S. Cuoco, Keng-Chi Chang, Gertrude Ecklu-Mensah, Aleena K. S. Arakaki, Noorsher Ahmed, Pratibha Jagannatha.

**Methodology:** Evan A. Boyle, Gabriela Goldberg.

**Project administration:** Evan A. Boyle, Gabriela Goldberg, Jonathan C. Schmok, Jillybeth Burgado, Fabiana Izidro Layng, Hannah A. Grunwald, Michael S. Cuoco, Gertrude Ecklu-Mensah, Ximena Garcia Arceo, Pratibha Jagannatha, Jonathan Pekar, Mallika Iyer.

**Resources:** Gene W. Yeo.

**Supervision:** Evan A. Boyle, Hannah A. Grunwald, Gene W. Yeo.

**Validation:** Evan A. Boyle, Keng-Chi Chang.

**Visualization:** Evan A. Boyle.

**Writing – original draft:** Evan A. Boyle, Gabriela Goldberg, Jonathan C. Schmok, Jillybeth Burgado, Gertrude Ecklu-Mensah.

**Writing – review & editing:** Evan A. Boyle, Gabriela Goldberg, Gene W. Yeo.

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
