## [Decision Letter · Decision Letter 0]

27 Apr 2023

PONE-D-22-17841Junior scientists spotlight social bonds in seminars for diversity, equity, and inclusion in STEMPLOS ONE

Dear Dr. Boyle,

Thank you for submitting your manuscript to PLOS ONE. I sincerely apologise for the unusually delayed review timeframe. Your manuscript has been assessed by two reviewers, whose comments are appended below. The reviewers comment positively about the study design and research question, but raise concerns regarding aspects of the methodology and discussion of the results that should be addressed. After careful consideration, we feel that it has merit but does not fully meet PLOS ONE’s publication criteria as it currently stands. However, we invite you to submit a revised version of the manuscript that addresses the points raised during the review process.

We look forward to receiving your revised manuscript.

Kind regards,

Emily Chenette

Editor in Chief

PLOS ONE

“DASL is funded in part by a grant from the Chan-Zuckerberg Initiative. E.A.B. acknowledges funding from the Helen Hay Whitney Foundation. G.W.Y. is an Allen Distinguished Investigator, a Paul G. Allen Frontiers Group advised program of the Paul G. Allen Family Foundation. The 2020 DASL symposium was supported by HHMI Gilliam Fellows, the ThermoFisher Queer Working Group, the Huntsman Cancer Institute, the UCSD Biology Student Social Fund, UCSD Graduate Division, UCSD Biomedical Sciences Graduate Program, the UCSD Biology Diversity Committee, the UCSD Office for Equity, Diversity, and Inclusion, the UCSD Latinx/Chicanx Academic Excellence Initiative, the UCSD Black Academic Excellence Initiative, and the UCSD School of Medicine. We thank the oSTEM UCSD graduate chapter leadership for hosting SQUAD and making seminars available on their website”

“DASL is funded in part by a grant from the Chan-Zuckerberg Initiative. E.A.B. acknowledges funding from the Helen Hay Whitney Foundation. G.W.Y. is an Allen Distinguished Investigator, a Paul G. Allen Frontiers Group advised program of the Paul G. Allen Family Foundation.

3. One of the noted authors is a group or consortium DASL Alliance. In addition to naming the author group, please list the individual authors and affiliations within this group in the acknowledgments section of your manuscript. Please also indicate clearly a lead author for this group along with a contact email address.

4. We note that Figure 1 in your submission contain [map/satellite] image which may be copyrighted. All PLOS content is published under the Creative Commons Attribution License (CC BY 4.0), which means that the manuscript, images, and Supporting Information files will be freely available online, and any third party is permitted to access, download, copy, distribute, and use these materials in any way, even commercially, with proper attribution. For these reasons, we cannot publish previously copyrighted maps or satellite images created using proprietary data, such as Google software (Google Maps, Street View, and Earth). For more information, see our copyright guidelines: http://journals.plos.org/plosone/s/licenses-and-copyright.

Reviewers' comments:

Reviewer's Responses to Questions

**Comments to the Author**

1. Is the manuscript technically sound, and do the data support the conclusions?

Reviewer #1: Partly

Reviewer #2: Partly

2. Has the statistical analysis been performed appropriately and rigorously? 

Reviewer #1: Yes

Reviewer #2: Yes

3. Have the authors made all data underlying the findings in their manuscript fully available?

Reviewer #1: No

Reviewer #2: Yes

4. Is the manuscript presented in an intelligible fashion and written in standard English?

Reviewer #1: Yes

Reviewer #2: Yes

5. Review Comments to the Author

Reviewer #1: The authors launched a weekly seminar in June 2020 called the Diversity and Science Lecture (DASL) series, where junior life scientists were invited to speak about their research and broader issues of diversity, equity, and inclusion in STEM. After a year, 79 speakers had participated, and the authors performed a descriptive analysis of the topics mentioned by speakers, and how those topics correlated with demographic and professional characteristics of the speakers. The major finding, as described in the title, is that almost every speaker talked about interpersonal support, but the authors describe many other patterns observed in the data as well.

This sentence, which had its own paragraph in the introduction, serves as a good summary of the main research question: "Who claims the opportunity afforded by DASL, and what are speakers’ priorities for equity, diversity, and inclusion in STEM?"

The study generally meets the criteria laid out by PLOS for publication, assuming the authors can address some methodological points and one minor question about data availability:

1. The paper could speak more to the recruitment process. Since the speakers were recruited by the seminar organizers, it is unclear to me how much the patterns described by the paper were induced by organizer selection vs. a more representative sampling of preferences among speakers. The authors were careful not to generalize beyond their sample, but a more detailed description of the recruitment criteria, how successful recruitment efforts tended to be, and whether there were demographic patterns among which speakers were more easily recruited could help the reader understand the generalizability of the results.

2. It is strange to me that the authors chose to use name-based methods to infer the demographics of the speaker participants, considering how (a) the sample size is small enough that the statistical errors from these tools will hard to quantify, and (b) the actual participants were highly available to the authors insofar as the seminar organizers directly recruited the speakers. The authors write that they chose not to use self-report data because of the following logistical challenges: "what, if any, prefilled options to offer, and how to handle missing data, spelling errors, misunderstandings, and possible identifiability of participants if results are published". However, those challenges are precisely carried over to name-based methods:

2i. Which prefilled options to offer in a survey is the same dilemma as deciding which categories the name-based inference tool can identify. The authors use off-the-shelf tools, but that doesn't justify their use. In fact, since the authors use data such as the US Census for the ground truth of these statistical tools, isn't that an implicit endorsement of the way that the US Census chooses to collect that data?

2ii. Missing data is not bypassed using name-based tools, since there are names that will fail to resolve, as the authors point out.

2iii. Spelling errors will also affect the performance of a name-based tool.

2iv. It is unclear how a name-based tool offers greater privacy or anonymity than a survey.

3. Moreover, the use of name-based methods is not mutually exclusive with a survey. Participants who do not report their demographic information in a survey could then be subject to a name-based inference to increase coverage, for example. If at all possible, I would like to see the authors complement their data with actual self-report demographics from participants, and if not, justify their decision to use name-based tools on different grounds than the ones they listed.

4. Reading the supplementary data file, it was unclear to me which of the keyword queries actually corresponded with the topic of "social factors". I'm assuming that's some combination of topics, but which ones?

5. The authors imply that the choice of topic by speaker is meaningful, writing: 'Topics related to social factors were most likely to be broadly discussed. “Education” keywords were most frequently mentioned (46/54 talks), followed by “Family” keywords (39/54) and “Mentoring” keywords (37/54).' It could be the case that these topics are mentioned simply because the queries associated with each given topic are broader or narrower, matching more or fewer words in the English language weighted by their popular usage. It would be more convincing for the authors to compare the rates that these keywords appear in the transcripts vs. some other English-language baseline (ideally a different life sciences seminar series that didn't focus on diversity, but even a comparison as different as Google n-grams or English Wikipedia would be better than nothing).

6. The discussion briefly mentions "unprecedented social distancing", since the seminars started in June 2020, but there is no explicit mention or analysis of COVID-related topics. Is that something that appears in the data?

7. The trainee speakers were given significantly shorter speaking times than the early career faculty. Can the authors describe in the paper how this affects (or doesn't affect) the different results across job classifications?

8. No need to act on this one, but just a thought: The data from the text transcripts is very rich, and there is something interesting about the temporal layout of a talk, and which topics lead to another.

9. The paper does not describe why the transcripts were not made available. Could they be? If not, why?

Reviewer #2: #

# Overview

#

This paper presents a study of the Diversity and Science Lecture Series (DASL), a speaker series began at UC San Diego in 2020. This series provides a venue for junior life scientists to share their research while also reflecting on their personal background and experiences with diversity, equity, & inclusion, and professional advice for their colleagues. This study investigates the gender, ethnic, and professional status of the speakers at DASL. Moreover, the authors analyze the transcripts of talks to quantify common themes and topics throughout the speaker series. Through these analyses, the authors observe that DASL is a series with diverse speakers, and that topics covered often reflect interpersonal support and their experience as member of a marginalized group.

On a technical level, the paper is fine. The methods is straightforward and seem properly implemented. Statistical tests are appropriate. Overall, the analysis is carried out competently. Figures are aesthetically pleasing, through some changes could make them more clear. The writing, too, is clear at the level of individual sentences. For these, I commend the authors.

Where the paper falls short is in its narrative. The individual elements of the study are competent, but they are not integrated into a coherent argument. In short, the authors do not do a good job of answering the question of “so what?” which is disappointing given that the speaker series itself seems incredibly valuable and the topic of fostering more inclusive environments vitally important. To summarize my concerns, the authors do not adequately motivate their study nor reason about the significance of their findings. I provide a more detailed accounting for how the paper might be changed below in a series of major comments, along with a series of minor comments that can be more easily addressed.

#

# Major comment 1: Motivation

#

The authors do not adequately motivate their study. To illustrate, I’ve done my best to outline the outline the structure of the argument made in the introduction of their study:

(1) The authors detail inequities in science

(2) It is therefore necessary to listen to marginalized voices

(3) DASL offers a venue for marginalized voices

(4) …

(5) The authors study representation and topics of DASL

(6) ...

I’ve added eclipses (…) for those stages of the argument that I feel are critical, and missing. Namely, I see the importance of DASL as a speaker series, but it is not made clear why it is important to study DASL (the missing stage 4). Neither is it clear what the desired benefit of studying DASL is (the missing stage 5); do the authors hope to understand something about speaker series in general, about marginalized groups, about DEI initiatives, do they hope to inform the management of other speaker series?

These missing stages are vital, because they determine why a casual reader, who may have likely never heard of DASL, should care.

I suggest the authors fill in these missing argumentative steps in their introduction, with the guiding goal of explaining to a casual reader (perhaps a social scientist, or a biologist interested in hosting their own speaker series) should be interested in this work.

#

# Major Comment 2: Discussion of significance of results

#

Building on the previous comment, the current discussion section is somewhat disjointed. There are many statements that more or less rise the evidence (see the next comment), but they are lacking in a coherent argument. Namely, there is not a clear argument about the significance of the study’s findings. To illustrate, below I highlight some of the paper’s findings:

- DASL is internationally diverse (more than another conference, but the baseline is confusing)

- DASL covered a broad range of topics relating to identify, racism, parenthood, mental health, etc.,

- There are notable correlations between certain topics, indicating the salience of intersectional identities

These findings are stated, more or less clearly, but little is done to contextualize the findings against policy, practice, the past literature, or theory. Simply put, why should a reader care?

Better motivating the study (as in comment #1) would make the significance more immediately clear, but the authors should also consider making a clear argument for the significance of their findings in the discussion. There are many narrative paths that they could take. The authors could discuss how their arguments can be understood in terms of established sociological theory. They might also consider discussing how their findings might inform future DEI policies and best practices. The paper could instead argue how these findings could inspire changes to other speaker series in order to make them more equitable and inclusive. Regardless of which narrative is chosen, making a more clear argument for the significance of the results could greatly increase the impact of this paper.

#

# Major comment 3: Claims in the existing discussion section

#

Several claims made in the current discussion section are poorly supported, which I feel stems from a lack of precision in language, clarity of the manuscript, or over-interpretation of results. I’ve provided a few examples below which highlight some of my concerns, which apply to other statements throughout the discussion.

- “We also exploited a geospatial analysis of speaker surnames that showed particular enthusiasm from Latino and Hispanic scientists.”

The authors use the word “enthusiasm”, but I don’t feel that the term can be used without a clearer understanding of how speaker slots in DASL were assigned. For example, if speakers had to be invited, then it could simply be that more Latino/Hispanic scholars were invited. It could also be that awareness of the series is not uniform, and that those from other groups (say, Asian scholars) would be more enthusiastic if they were made aware. This is all to say that there can be differential outcomes in who speaks at DASL even if we assume that every person has equal enthusiasm. I urge the authors to explain how speakers are chosen and how DASL was advertised. Additionally, it may be worthwhile to revisit this phrase.

- “It is perhaps no surprise that unprecedented social distancing measures prompted trainee life scientists to reflect on the contribution of interpersonal relationships to equity, diversity, and inclusion in STEM””

The authors do not, as far as I can tell, have any results that explain differentials in topics covered pre- and during-covid. Therefore, claims about the pandemic affecting speaker’s interpersonal relationships, while almost certainly true, cannot be supported by the evidence that is shown to the reader. I urge the authors to reconsider whether conclusions drawn in this section are adequately supported by evidence in the manuscript.

- “Latino/Hispanic individuals and those from rural backgrounds especially prioritize family”

This is a statement that is partially supported by the result, but thin to the point that I would further qualify it. Figure 4.b shows a correlation in the terms in talk transcripts that relate to Latino/Hispanic identify & Family. What 4.b does not show is that Latino/Hispanic speakers, themselves, “prioritize” family to a greater extent than other groups considered. What is missing to make this claim is evidence that a) Latino/Hispanic speakers are those using words relating to Latino/Hispanic identify (very likely, but need to be sure), and b) that mentions made to words like “family” reflect a prioritization of family, whatever is meant by the term “prioritization”. I suggest the authors rewrite this sentences, and others like it in the discussion, to be a bit more precise, which risks inviting skepticism or criticism from a reader.

#

# Major Comment 4: Justifying baselines

#

The authors conduct several comparisons between their data and other speaker series, which I refer to as baselines. However, it is not clear why these baselines are chosen or what insights, exactly, comparison against this baseline provides.

The first baseline involves examining the ethnic breakdown of speakers of the Fragile Nucleosome (FN) speaker series. This baseline, as far as I can tell, is included to support the argument that DASL is more diverse than peer events. However, it is not at all clear that FN is a peer speaker series. Whereas DASL seems to have emerged as UC San Diego, the FN appears to be an international-by-default organization, is not explicitly motivated by DEI goals, reflects a specific subspecialty of biology, and does not appear oriented primarily towards graduate students. My question then, is what is the significance of comparing DASL to FN?

The second baseline is the Scientific Queers United in Academic Discourse (SQUAD) series. SQUAD appears a closer to a peer event to DASL, but it is still not clear what is accomplished by comparing the two. I can envision SQUAD being used to support the generalizability of results from analyzing DASL. For example, that SQUAD has a focus on friendship suggests that perhaps this is a more general finding across different contexts, whereas the greater mention of “Asian” as an identify reflects a deficit in DASL. Regardless of what the significance is, it should be more clearly stated.

I urge the authors to consider what these baseline comparisons can tell the reader about DASL. At present, Fragile Nucleosome and SQUAD are not mentioned in the discussion, and nothing is done to justify why these particular speaker series are selected.

#

# Minor comments

#

Below I have a series of comments, mostly minor, that should be more easily implemented. First, I list comments for each figure, then a seres of of other miscellaneous comments.

- Figure 1: It’s not clear what the colors of points mean, exactly. Either visual cues should be made more specific, a legend added, or the definitions of each color made more explicit in the caption.

- Figure 2: What does the color mean in the word clouds? Does it correspond to the colors used in panels (b) and (c)? If so, this should be stated explicitly. If not, then colors should be removed or at least made different than in the other panels. Following on this, it is confusing to use the same colors to reflect different categories in panels (b) and (c), I suggest removing color entirely, or using a consistent palette to reflect the same categories throughout the paper (e.g., “Graduate Student” should always appear orange).

- Figure 3: Panel (c): the legend direction should be reversed, starting with “7” on the left is a bit confusing.

- Figure 4: I’m not convinced that the heat map-style approach in panel (a) is a good way to present these data, its simply too difficult to make any inferences. Honestly, a bar graph simply showing the number to talks mentioning each keyword family would be much more helpful.. Panel (c): What exactly is going on with the infinite odds rational dn why is the student-speaker “Asian” keyword marked here? Also, is this panel showing the most disproportionately used keyword? Its not entirely clear.

- The authors define categories of keywords, such as “Family” and “Education”, but it is not clear where these categories come from. Were they drawn from past literature? Or were they originally defined by the authors? If the authors created them, then how were they chosen? Moreover, the reader would benefit from listing some of the example keywords of each category (or at least a few of the major categories) in the text itself.

- In the section “DASL speakers emphasize the importance of social and interpersonal factors in STEM”, the authors suddenly begin examining keywords across 54 talks, but the number of talks previously stated was 79. Its explained later that only a subset of talks are associated with transcripts, but this should really be made more clear in this section.

- When conducting statistical tests, it is generally good practice to list effect sizes immediately alongside the p-value.

- Generally, social science literature places the methodology just after the introduction and before the results section. Readers often like to gain a sense of the data before approaching the result. While not strictly necessary, it could be beneficial for the authors to follow this standard.

- Its confusing that figure 5 is mentioned before figure 4.c,d,e. Either the result text or the panels themselves should be re-arranged

- The results of Figure 4d and 4e don’t appear to be mentioned in the discussion section. They deserve to be incorporated into the argument of the paper.

6. PLOS authors have the option to publish the peer review history of their article (what does this mean?). If published, this will include your full peer review and any attached files.

Reviewer #1: No

Reviewer #2: No

---

## [Author Response · Author response to Decision Letter 0]

11 Jun 2023

Reviewer #1: The authors launched a weekly seminar in June 2020 called the Diversity and Science Lecture (DASL) series, where junior life scientists were invited to speak about their research and broader issues of diversity, equity, and inclusion in STEM. After a year, 79 speakers had participated, and the authors performed a descriptive analysis of the topics mentioned by speakers, and how those topics correlated with demographic and professional characteristics of the speakers. The major finding, as described in the title, is that almost every speaker talked about interpersonal support, but the authors describe many other patterns observed in the data as well.

This sentence, which had its own paragraph in the introduction, serves as a good summary of the main research question: "Who claims the opportunity afforded by DASL, and what are speakers’ priorities for equity, diversity, and inclusion in STEM?"

The study generally meets the criteria laid out by PLOS for publication, assuming the authors can address some methodological points and one minor question about data availability:

Boyle: We are heartened by the reviewer’s positive response to our work. We believe our peers will greatly benefit in their efforts to run DEI programming and implement impactful change from our revised work, which is more transparent with respect to underlying data and more holistic given the incorporation of self-reported data.

R1.1. The paper could speak more to the recruitment process. Since the speakers were recruited by the seminar organizers, it is unclear to me how much the patterns described by the paper were induced by organizer selection vs. a more representative sampling of preferences among speakers. The authors were careful not to generalize beyond their sample, but a more detailed description of the recruitment criteria, how successful recruitment efforts tended to be, and whether there were demographic patterns among which speakers were more easily recruited could help the reader understand the generalizability of the results.

Boyle: We agree with the reviewer that the recruitment process could change the interpretation of our results substantially. We have added information on how speakers were recruited: see below. We encouraged nominations for anyone who might want to participate in DASL for any reason. It’s possible that the organizers’ selections for early speakers set an unstated precedent that continued for the rest of the series. The volume of talks has scaled to meet the demand for the platform so we have been able to accommodate almost every nominated speaker.

Summer quarter DASL speakers and all faculty speakers were recruited directly by DASL organizers representing the the graduate programs in Biological Sciences, Biomedical Sciences, Bioinformatics and Systems Biology, Bioengineering, and Neurosciences, plus postdoctoral scholars appointed in UC San Diego’s School of Medicine and School of Biological Sciences. In subsequent quarters, nominations were accepted through a Google Form posted on the DASL website and shared on the DASL email list. Nominated trainee scientists were contacted and scheduled if the DASL platform appealed to them.

We added a sentence to the Discussion to emphasize the role that recruitment might play in our results.

Attention paid to each topic is likely influenced by the seminar organizers involved and the employed recruitment strategy.

R1.2. It is strange to me that the authors chose to use name-based methods to infer the demographics of the speaker participants, considering how (a) the sample size is small enough that the statistical errors from these tools will hard to quantify, and (b) the actual participants were highly available to the authors insofar as the seminar organizers directly recruited the speakers. The authors write that they chose not to use self-report data because of the following logistical challenges: "what, if any, prefilled options to offer, and how to handle missing data, spelling errors, misunderstandings, and possible identifiability of participants if results are published". However, those challenges are precisely carried over to name-based methods:

2i. Which prefilled options to offer in a survey is the same dilemma as deciding which categories the name-based inference tool can identify. The authors use off-the-shelf tools, but that doesn't justify their use. In fact, since the authors use data such as the US Census for the ground truth of these statistical tools, isn't that an implicit endorsement of the way that the US Census chooses to collect that data?

2ii. Missing data is not bypassed using name-based tools, since there are names that will fail to resolve, as the authors point out.

2iii. Spelling errors will also affect the performance of a name-based tool.

2iv. It is unclear how a name-based tool offers greater privacy or anonymity than a survey.

Boyle: As part of its efforts to improve the visibility of its speakers and facilitate recruitment of diverse speakers, DASL surveyed past speakers and posted the results to its website.

https://www.ucsddasl.com/speakers

We downloaded the survey results from the website. We found information on self-identified race, other identities, talk themes, and interests for 31 speakers from the first year of DASL and 5 from the second. We now summarize self-reported identities and themes:

To summarize DASL speakers and talk content, we analyzed the identities and motives of 36 DASL speakers posted on the DASL website as well as the themes each speaker chose to describe their talk (Supplementary Figure 4). First generation status was claimed by nearly half of the speakers (17/36, Supplementary Figure 4a). Low socioeconomic status, immigrant identity, LGBTQ+ identity, and rural and agriculture background were each claimed by approximately a quarter of the speakers. An overwhelming majority of speakers (33/36) wanted to contribute to discourse on DEI, and nearly two-thirds (23/36) selected it as the top reason. In descending order, mentorship, women in science, and the importance of education were the most commonly chosen themes. Especially salient themes included women in science, sexual and gender minorities, and rural and agricultural background, each of which were the top theme for half of speakers who chose it. 

We also tested for co-occurrence across 23 identities, motives, and themes listed by DASL speakers and called 39 correlated or anticorrelated responses (permutation of spearman correlation coefficient, 10% FDR, Supplementary Figure 4b). Some pairing of identities and themes were wholly predictable: woman identity to the theme of women in science, LGBTQ+ identity to the theme of sexual and gender minorities, rural identity to the theme of rural and agriculture background, and low socioeconomic status to the theme of financial barriers. Some co-occurring identities pointed to racial disparities: low socioeconomic status with Hispanic/Latino, American Indian/Alaska Native, and multiracial identities, disability with American Indian/Alaska Native race, and mental health for white race. In some cases, co-occurrence suggested diverging attention across identities: Women speakers less often discussed sexual and gender minorities or rural and agricultural background, and Black speakers expressed more interest in DASL for the opportunity to present their research.

The self-identified data (see Supplementary Figure 3) appear to agree very well with our association approach. We added the results to the manuscript after the nationality accuracy benchmark:

We re-examined predictions using publicly declared nationalities from select speakers as well as racial identities of speakers posted on the DASL website. For the 14 speakers who publicly identified their nationality, we found a 71% success rate for lookup in Forebears versus 36% for LSTM predictions. LSTM predictions for Latin American speakers’ names were often linked to Anglophone regions (Figure 3c), suggesting that overrepresentation of contributors from Anglophone countries in Wikipedia may cause bias against Latin American names in the learned LSTM. On the DASL website, two speakers preferred not to disclose their race, but 29 out of 79 speakers reported their racial identity on the DASL website (Supplementary Figure 3). Of those, 1 was unidentified and 26 had a matching associated region or census-based inference (93% accuracy). Of note, no speakers associated to the Middle East or North Africa reported data to the DASL website. Surname association by direct lookup with racial disambiguation by census data excelled in annotating all speakers’ backgrounds.

Interestingly, while the response rate obviously limits the completeness of the dataset, it appears that the number of speakers who would prefer not to disclose race (even if they were to complete the survey) exceeds the number of speakers who cannot be identified by direct lookup. All things considered, missing data is a far greater challenge for studies based on self identification data than for inference.

 The DASL organizers received a complaint from a past participant who did not want to be bothered with survey reminders, and surname inference can be done without storing any personal contact information, so we believe there are privacy benefits, if modest. 

Programmatic and semi-automated methods that make inferences based on names or documents have the potential to curtail response bias, protect participant privacy (e.g., no exposure of phone number, mailing address, or email address), and rescue data lacking self-identification

In any case, the survey results make it possible to address an important area of identity that was wholly inaccessible from our name based approach: multi-racial (or multi-national) identity. In this respect self-identification is essential.

R1.3. Moreover, the use of name-based methods is not mutually exclusive with a survey. Participants who do not report their demographic information in a survey could then be subject to a name-based inference to increase coverage, for example. If at all possible, I would like to see the authors complement their data with actual self-report demographics from participants, and if not, justify their decision to use name-based tools on different grounds than the ones they listed.

Boyle: We have added more analysis based on posted DASL survey results. The differences between surname inference and self identification naturally posed some challenges to integrating the two, but it is clear that the profile and transcript-only approach benefits tremendously from scaling to all 54 recorded talks instead of the 18 with corresponding survey results (13 of 31 respondents were not recorded or declined to share). Even with a more lenient 20% FDR, only 7 nontrivial intersectional identities could be retrieved compared to approximately 19 (depending on how one decides trivial identities) by analyzing the transcripts. The transcript approach is also aided by the quantitative signal in the number of mentions as opposed to the discrete nature of identifying as a race or another identity.

R1.4. Reading the supplementary data file, it was unclear to me which of the keyword queries actually corresponded with the topic of "social factors". I'm assuming that's some combination of topics, but which ones?

Boyle: Keyword groups are shown in Figure 4 on the right. The analyses do not rely heavily on the key word groups, which are quite subjective. The keywords belonging to “Education”, “Family”, “Mentoring”, “Friends”, “First generation”, and “Class” are grouped under “social factors”. Finances and COVID-19 are arguably also social factors but are classified under concepts.

The topics and keywords are now available in the Supplementary data so that every occurrence of a term can be reviewed.

R1.5. The authors imply that the choice of topic by speaker is meaningful, writing: 'Topics related to social factors were most likely to be broadly discussed. “Education” keywords were most frequently mentioned (46/54 talks), followed by “Family” keywords (39/54) and “Mentoring” keywords (37/54).' It could be the case that these topics are mentioned simply because the queries associated with each given topic are broader or narrower, matching more or fewer words in the English language weighted by their popular usage. It would be more convincing for the authors to compare the rates that these keywords appear in the transcripts vs. some other English-language baseline (ideally a different life sciences seminar series that didn't focus on diversity, but even a comparison as different as Google n-grams or English Wikipedia would be better than nothing).

Boyle: We were unable to find a suitable matched sample of words. However, we include a summary of self-reported priorities from DASL speakers from the DASL website. We find that education and mentorship are two of the top three topics for both self-reported talk themes and transcript inference. The self-reported and inferred themes/topics (as well as the race/nationality inference) appear quite consistent.

R1.6. The discussion briefly mentions "unprecedented social distancing", since the seminars started in June 2020, but there is no explicit mention or analysis of COVID-related topics. Is that something that appears in the data?

Boyle: Most speakers did not address COVID-19 explicitly in their talks, but mentions were not uncommon. We have now included COVID-19 in the keyword analysis. We report that speakers who mentioned keywords assigned to sexual and gender minorities or first generation status were more likely to mention COVID-19. 

Mentions of “First generation” correlated with “COVID-19”, which suggests that disruptions from the pandemic to higher education might be particularly salient for those who are first in their family to attend. These relationships were not evident from the limited data speakers self-reported on the DASL website.

R1.7. The trainee speakers were given significantly shorter speaking times than the early career faculty. Can the authors describe in the paper how this affects (or doesn't affect) the different results across job classifications?

Boyle: None of the early career faculty allowed recording of their talk so they were only included in the profile analyses - we expect these are not affected by speaking time differences.

R1.8. No need to act on this one, but just a thought: The data from the text transcripts is very rich, and there is something interesting about the temporal layout of a talk, and which topics lead to another.

Boyle: We completely agree that the temporal information from the transcript timestamps is worth investigating. We drafted preliminary figures pursuing this angle but the flexibility of the DASL talk format made it challenging: speakers varied in whether they included DEI content at the beginning or end of their talk or interspersed throughout. Tracks of mentions usually included long blank stretches. The reviewer may have a point about a simpler approach on investigating the ordering without worrying about the absolute time between mentions.

R1.9. The paper does not describe why the transcripts were not made available. Could they be? If not, why?

Boyle: We have asked the speakers if they would like to make their talks public. We are happy to provide the transcripts to anyone who wants to follow up on our analyses, but at the time of presentation, many talks contained confidential unpublished research. We considered censoring the research parts of the talk but it was somewhat subjective, and preliminary efforts were frustratingly slow. To improve access to the data and the precise words being matched, we have included precise words and their surrounding context. We included the point in the transcript of the occurrence that would permit some basic temporal analysis.

DASL organizers have received permission to post several talks to its YouTube channel. 3rd party websites can be used to download the talks and analyze the transcripts. We hope that more speakers will grant permission to share their talks publicly over time.

See DASL’s channel here: https://www.youtube.com/channel/UCMPW5OEZKL_90C2KgARFJ3Q

Reviewer #2: #

# Overview

#

This paper presents a study of the Diversity and Science Lecture Series (DASL), a speaker series began at UC San Diego in 2020. This series provides a venue for junior life scientists to share their research while also reflecting on their personal background and experiences with diversity, equity, & inclusion, and professional advice for their colleagues. This study investigates the gender, ethnic, and professional status of the speakers at DASL. Moreover, the authors analyze the transcripts of talks to quantify common themes and topics throughout the speaker series. Through these analyses, the authors observe that DASL is a series with diverse speakers, and that topics covered often reflect interpersonal support and their experience as member of a marginalized group.

On a technical level, the paper is fine. The methods is straightforward and seem properly implemented. Statistical tests are appropriate. Overall, the analysis is carried out competently. Figures are aesthetically pleasing, through some changes could make them more clear. The writing, too, is clear at the level of individual sentences. For these, I commend the authors.

Where the paper falls short is in its narrative. The individual elements of the study are competent, but they are not integrated into a coherent argument. In short, the authors do not do a good job of answering the question of “so what?” which is disappointing given that the speaker series itself seems incredibly valuable and the topic of fostering more inclusive environments vitally important. To summarize my concerns, the authors do not adequately motivate their study nor reason about the significance of their findings. I provide a more detailed accounting for how the paper might be changed below in a series of major comments, along with a series of minor comments that can be more easily addressed.

Boyle: We truly appreciate the reviewer’s commitment to understanding what we accomplished with DASL and maximizing the impact of the manuscript. We have added roughly two paragraphs of content to the Introduction to expand upon the motivation for the work and thoroughly revised the Discussion to provide more specific examples of what we learned focus on three main points: methods for computational audits, what we learned about speaker and organizer recruitment for DEI prograrmming, and how the discovery of intersectional identities and differential topics of concern might translate to policy decisions. We hope the reviewer finds that numerous analyses added in revision provide further credence to our claims.

#

# Major comment 1: Motivation

#

The authors do not adequately motivate their study. To illustrate, I’ve done my best to outline the outline the structure of the argument made in the introduction of their study:

(1) The authors detail inequities in science

(2) It is therefore necessary to listen to marginalized voices

(3) DASL offers a venue for marginalized voices

(4) …

(5) The authors study representation and topics of DASL

(6) ...

I’ve added eclipses (…) for those stages of the argument that I feel are critical, and missing. Namely, I see the importance of DASL as a speaker series, but it is not made clear why it is important to study DASL (the missing stage 4). Neither is it clear what the desired benefit of studying DASL is (the missing stage 5); do the authors hope to understand something about speaker series in general, about marginalized groups, about DEI initiatives, do they hope to inform the management of other speaker series?

These missing stages are vital, because they determine why a casual reader, who may have likely never heard of DASL, should care.

I suggest the authors fill in these missing argumentative steps in their introduction, with the guiding goal of explaining to a casual reader (perhaps a social scientist, or a biologist interested in hosting their own speaker series) should be interested in this work.

Boyle: We revised the introduction to fill in the 4th bullet point as recommended by the reviewer.

Who claims the opportunity afforded by a new diversity-themed seminar series like DASL, and what are speakers’ priorities for equity, diversity, and inclusion in STEM? 

Answering such questions can inform further efforts to build platforms for DEI in STEM and translate a transient awareness of DEI concerns into enduring policy changes. An understanding of what candidate speakers value and which facets of diversity may be absent greatly facilitates speaker recruitment. After hosting speakers, documentation of recurrent concerns and recommended remedies can point to both areas in need of restructuring or programs meriting expansion.

Yet, seminars typically consist of unstructured speech. Coding speaker profiles and speech into quantitative data entails tremendous discretion and diminishes speakers’ qualitative experiences, but basing decisions on qualitative impressions makes effective prioritization of goals difficult and risks inaction on comparatively important but less salient concerns. 

We further revised the end of the introduction to fill in the 6th bullet point as recommended by the reviewer.

Here, we summarize how trainee life scientists conceive of equity, diversity, and inclusion in STEM (Figure 1b). We find large differences in representation compared to a

traditional seminar series and identify differences in topic noteworthiness that vary by speaker background and priorities. We quantify self-reported reasons junior life scientists give to participate in a new diversity-themed seminar series as well as the topics most often discussed as assessed from seminar recordings. We review resources that DASL speakers cited as personally helpful and characterize how different minority populations may be best served by different types of DEI programming. Our work provides a template for future computational audits of diversity in STEM that provide actionable insight for speaker recruitment and priorities for DEI policy changes.

#

# Major Comment 2: Discussion of significance of results

#

Building on the previous comment, the current discussion section is somewhat disjointed. There are many statements that more or less rise the evidence (see the next comment), but they are lacking in a coherent argument. Namely, there is not a clear argument about the significance of the study’s findings. To illustrate, below I highlight some of the paper’s findings:

- DASL is internationally diverse (more than another conference, but the baseline is confusing)

- DASL covered a broad range of topics relating to identify, racism, parenthood, mental health, etc.,

- There are notable correlations between certain topics, indicating the salience of intersectional identities

These findings are stated, more or less clearly, but little is done to contextualize the findings against policy, practice, the past literature, or theory. Simply put, why should a reader care?

Better motivating the study (as in comment #1) would make the significance more immediately clear, but the authors should also consider making a clear argument for the significance of their findings in the discussion. There are many narrative paths that they could take. The authors could discuss how their arguments can be understood in terms of established sociological theory. They might also consider discussing how their findings might inform future DEI policies and best practices. The paper could instead argue how these findings could inspire changes to other speaker series in order to make them more equitable and inclusive. Regardless of which narrative is chosen, making a more clear argument for the significance of the results could greatly increase the impact of this paper.

Boyle: We thank the reviewer for their suggestions on the Discussion. We have thoroughly rearranged and rewritten the entire Discussion in response. We focused on the major points centering the work: the methods for performing computational audits, the insight into speaker recruitment, and what we learned that could inform policy or funding decisions. We hope the reviewer finds the narrative more streamlined and digestible.

Example of new text added:

Interestingly, speakers pointed primarily to national sources of funding as personally helpful, suggesting a disconnect between the fungible nature of funding and the interpersonal support desired by trainee life scientists (Figure 4d). We believe that trainee life scientists highly esteem a sense of camaraderie and that access to interpersonal support is essential for achieving equity, diversity, and inclusion at academic institutions even when funding levels are ample.

The visibility granted by these concerns at DASL have direct policy implications: LGBTQ+ and disabled identifying groups may benefit from more access to community space and funding for social events whereas Latino/Hispanic identifying groups may benefit from investing in child care subsidies and family leave. An overly narrow approach to DEI programming may leave certain minority groups dissatisfied given their unique priorities.

#

# Major comment 3: Claims in the existing discussion section

#

Several claims made in the current discussion section are poorly supported, which I feel stems from a lack of precision in language, clarity of the manuscript, or over-interpretation of results. I’ve provided a few examples below which highlight some of my concerns, which apply to other statements throughout the discussion.

- “We also exploited a geospatial analysis of speaker surnames that showed particular enthusiasm from Latino and Hispanic scientists.”

The authors use the word “enthusiasm”, but I don’t feel that the term can be used without a clearer understanding of how speaker slots in DASL were assigned. For example, if speakers had to be invited, then it could simply be that more Latino/Hispanic scholars were invited. It could also be that awareness of the series is not uniform, and that those from other groups (say, Asian scholars) would be more enthusiastic if they were made aware. This is all to say that there can be differential outcomes in who speaks at DASL even if we assume that every person has equal enthusiasm. I urge the authors to explain how speakers are chosen and how DASL was advertised. Additionally, it may be worthwhile to revisit this phrase.

Boyle: We completely agree with the reviewer that ‘enthusiasm’ is not the right word choice. We have rewritten the text.

We also exploited a geospatial analysis of speaker surnames that showed strong engagement with Latino and Hispanic scientists.

Reviewer 1 also noted that details on speaker recruitment were extremely important for proper interpretation. Over its history, DASL has had twenty organizers actively contribute and influence speaker recruitment. We have worked hard to accommodate every nominated speaker who has expressed interest. We added a subheading in the methods section to describe recruitment.

Speaker recruitment

Summer quarter DASL speakers and all faculty speakers were recruited directly by DASL organizers representing the the graduate programs in Biological Sciences, Biomedical Sciences, Bioinformatics and Systems Biology, Bioengineering, and Neurosciences, plus postdoctoral scholars appointed in UC San Diego’s School of Medicine and School of Biological Sciences. In subsequent quarters, nominations were accepted through a Google Form posted on the DASL website and shared on the DASL email list. Nominated trainee scientists were contacted and scheduled if the DASL platform appealed to them.

R2.3b - “It is perhaps no surprise that unprecedented social distancing measures prompted trainee life scientists to reflect on the contribution of interpersonal relationships to equity, diversity, and inclusion in STEM””

The authors do not, as far as I can tell, have any results that explain differentials in topics covered pre- and during-covid. Therefore, claims about the pandemic affecting speaker’s interpersonal relationships, while almost certainly true, cannot be supported by the evidence that is shown to the reader. I urge the authors to reconsider whether conclusions drawn in this section are adequately supported by evidence in the manuscript.

Boyle: We have rephrased the sentence in question to avoid claims of causation, which is outside the scope of the manuscript.

Junior life scientists’ reflections on the contribution of interpersonal relationships to equity, diversity, and inclusion in STEM were timely amidst the unprecedented social distancing measures implemented early in the COVID-19 pandemic. Still, it is notable that discussion of social factors including education and mentorship surpassed that of racial identity in a diversity-themed seminar series, as assessed by analyzing talk transcripts (Figure 5a) and the speakers' self-reported intent (Supplementary Figure 4a).

R2.3c- “Latino/Hispanic individuals and those from rural backgrounds especially prioritize family”

This is a statement that is partially supported by the result, but thin to the point that I would further qualify it. Figure 4.b shows a correlation in the terms in talk transcripts that relate to Latino/Hispanic identify & Family. What 4.b does not show is that Latino/Hispanic speakers, themselves, “prioritize” family to a greater extent than other groups considered. What is missing to make this claim is evidence that a) Latino/Hispanic speakers are those using words relating to Latino/Hispanic identify (very likely, but need to be sure), and b) that mentions made to words like “family” reflect a prioritization of family, whatever is meant by the term “prioritization”. I suggest the authors rewrite this sentences, and others like it in the discussion, to be a bit more precise, which risks inviting skepticism or criticism from a reader.

Boyle:

a) We indeed prove from speaker self-identified data that Hispanic/Latino identifying speakers mention Hispanic/Latino keywords. See new Figure 5b: 

Topics appeared to vary in breadth and salience across DASL talks (Figure 5a). To confirm that the correlations we reported were not an artifact of common words or misassignment of topics from keywords, we performed a similar transcript-based analysis for the 18 talks with corresponding self-identified data from the speaker on the DASL website using a relaxed 20% FDR threshold. We detected 16 topics that varied across speaker race, other identity, or interest or talk theme and found strong evidence that talk content directly reflects speaker identity: Hispanic/Latino and Asian speakers more often mentioned their respective races, immigrant, LGBTQ+, and rural background speakers more often mentioned their respective identities, and low socioeconomic status identifying speakers more often mentioned financial concepts (Figure 5b). We confirmed intersectional concerns for immigrant speakers discussing Asian identity, and LGBTQ+ speakers discussing male identity, giving credence to the transcript-only results reported above.

b) The indicated statements in the Discussion are speculative and far from proven. We have taken the reviewer’s advice and edited the text to avoid overstating our point.

We hypothesize that sexual and gender minorities and people with disabilities on average pay greater attention to friendship in order to connect with those with a shared identity. By comparison, Latino/Hispanic individuals and those from rural backgrounds on average pay greater attention to family, who likely share in those identities. 

#

# Major Comment 4: Justifying baselines

#

The authors conduct several comparisons between their data and other speaker series, which I refer to as baselines. However, it is not clear why these baselines are chosen or what insights, exactly, comparison against this baseline provides.

The first baseline involves examining the ethnic breakdown of speakers of the Fragile Nucleosome (FN) speaker series. This baseline, as far as I can tell, is included to support the argument that DASL is more diverse than peer events. However, it is not at all clear that FN is a peer speaker series. Whereas DASL seems to have emerged as UC San Diego, the FN appears to be an international-by-default organization, is not explicitly motivated by DEI goals, reflects a specific subspecialty of biology, and does not appear oriented primarily towards graduate students. My question then, is what is the significance of comparing DASL to FN?

The second baseline is the Scientific Queers United in Academic Discourse (SQUAD) series. SQUAD appears a closer to a peer event to DASL, but it is still not clear what is accomplished by comparing the two. I can envision SQUAD being used to support the generalizability of results from analyzing DASL. For example, that SQUAD has a focus on friendship suggests that perhaps this is a more general finding across different contexts, whereas the greater mention of “Asian” as an identify reflects a deficit in DASL. Regardless of what the significance is, it should be more clearly stated.

I urge the authors to consider what these baseline comparisons can tell the reader about DASL. At present, Fragile Nucleosome and SQUAD are not mentioned in the discussion, and nothing is done to justify why these particular speaker series are selected.

Boyle: We appreciate the reviewer’s interest in comparisons to FN and SQUAD. Comparison to FN largely confirmed that DASL was geospatially more diverse than a non-DEI concurrent remote life science series that featured >100 speakers. Comparison to SQUAD confirmed that different DEI seminars have different focuses, and further reinforced the underrepresentation of Asian speakers at DASL.

See this part of the Discussion:

We also exploited a geospatial analysis of speaker surnames that showed strong engagement with Latino and Hispanic scientists that far surpassed that seen in Fragile Nucleosome, a non-DEI life science seminar series (Supplementary Figure 2). Our tallies of keyword mentions in talk transcripts indicated that social factors like family, education, and mentorship are highly salient and broadly noteworthy (Figure 5a). Sexual and gender minority was noteworthy to fewer life science trainees but still highly salient. Our review of content recommended by speakers addressed offered more evidence for sexual and gender minority identity to select trainees (Figure 4e).

Our audit of speaker diversity yielded actionable insight into the operation of DASL. Men and Asian DASL speakers - especially Asian men speakers - were poorly represented. Only graduate students discussed Asian identity amongst recorded talks, suggesting the need for greater visibility of Asian postdoctoral scholars. Comparison to a SQUAD, a companion DEI series on LGBTQ+ scientists, further emphasized the dearth of Asian speakers at DASL (Figure 6). Men speakers disproportionately commented on finances, sexual and gender minorities, and rural and agricultural background, suggesting that a greater focus on these aspects of diversity may be a way to engage more men. In response to these results, DASL organizers intentionally recruited Asian and men speakers for its second year of seminars.

We further learned about effective speaker recruitment for diversity-themed seminar series. We found that contributing to discourse on DEI was highly motivating for junior life scientists (Supplementary Figure 4a). Furthermore, Speakers reported unanimous approval for dry runs that allowed them to receive feedback on their presentations before formal sessions. Based on DASL speakers’ self reported interest, we expect that similar programming focused on mentorship would achieve high engagement. Ultimately, disparities for women [19–23], racial minorities [8,24–27], and sexual and gender minorities [28–31] working in scientific disciplines have been well documented, and a seminar series offers a path for continued visibility.

#

# Minor comments

#

Below I have a series of comments, mostly minor, that should be more easily implemented. First, I list comments for each figure, then a seres of of other miscellaneous comments.

- Figure 1: It’s not clear what the colors of points mean, exactly. Either visual cues should be made more specific, a legend added, or the definitions of each color made more explicit in the caption.

Boyle: Figure 1 has been updated with an improved, originally produced World Map and legend specifying all informative colors.

Figure 1: Reviewing the first year of DASL seminars. a) A timeline of DASL milestones separated by quarter (yellow = summer, red = fall, blue = winter, green = spring). DASL was founded in June 2020 and completed its recent spring series in June 2021. b) Outline of the approach taken to synthesize insight from 79 weekly DASL seminar series speakers. World map colors specify geospatial regions (green = Europe, light green = Anglophone, yellow = Latin America, Spain, Portugal & the Philippines, red = Africa, purple = Middle East & North Africa, blue = Asia)

- Figure 2: What does the color mean in the word clouds? Does it correspond to the colors used in panels (b) and (c)? If so, this should be stated explicitly. If not, then colors should be removed or at least made different than in the other panels. Following on this, it is confusing to use the same colors to reflect different categories in panels (b) and (c), I suggest removing color entirely, or using a consistent palette to reflect the same categories throughout the paper (e.g., “Graduate Student” should always appear orange).

Boyle: The color in the word cloud redundantly encoded the frequency/size of each word. 

We have minimized instances where colors convey distinct and substantive information within the same figure. Each panel in figure 2 now has a single color to reduce any confusion.

- Figure 3: Panel (c): the legend direction should be reversed, starting with “7” on the left is a bit confusing.

Boyle: Great catch, we have updated the figure.

- Figure 4: I’m not convinced that the heat map-style approach in panel (a) is a good way to present these data, its simply too difficult to make any inferences. Honestly, a bar graph simply showing the number to talks mentioning each keyword family would be much more helpful.. Panel (c): What exactly is going on with the infinite odds rational dn why is the student-speaker “Asian” keyword marked here? Also, is this panel showing the most disproportionately used keyword? Its not entirely clear.

Boyle: We believe the heatmap helps illustrate how topics co-occur, as tested specifically in panel b. Nonetheless, we agree that summarizing the number of mentions per topic is important. This is shown in Figure 5a. 

We have retitled panel c to more clearly communicate that it is testing differential topic mentions by speaker identity as found in our speaker profiles (in practice this only tests gender and position held since we did not attempt stratification by talk title or research topic). No postdoc or professor mentioned Asian keywords - thus the odds ratio is infinite for student speakers. Only three identity-topic pairings were significant at a 10% FDR. New analyses of survey data (Supplementary Figure 4b, Figure 5b) showed the same co-occurring hits plus a few weaker hits for speaker-selected themes. Because women speakers were so likely to select women in science as one of three relevant themes, it is unclear whether the weaker hits are adding additional information.

- The authors define categories of keywords, such as “Family” and “Education”, but it is not clear where these categories come from. Were they drawn from past literature? Or were they originally defined by the authors? If the authors created them, then how were they chosen? Moreover, the reader would benefit from listing some of the example keywords of each category (or at least a few of the major categories) in the text itself.

Boyle: We have added a paragraph in the methods to describe how we managed topics and keywords, which was a collaborative and mostly ad hoc process informed by attending dozens of DASL talks.

Topics and keywords for quantitative content analysis were developed over multiple biweekly DASL organizer meetings. Over ten trainee life scientists from disparate graduate programs and departments participated and brainstormed topics. Draft results were shown and topics were added or subdivided per organizer feedback until the organizers were satisfied that the topics reflected the breadth of content covered in DASL seminars. For example, organizers decided to add a failure topic (keywords: struggle, challenge, fail, overwhelmed), and organized decided to divide socioeconomic divided into class (keywords: poor, poverty, income) and finances (keywords: pay, money, afford).

Boyle: We have made the transcript context for the keyword matches available in the supplementary data under “keyword matches.” There are instances of keywords giving a false topic match, but we implemented just a few basic safeguards in pattern matching, and the results should give a sense of the inaccuracies when processing text at scale where manual review is impractical.

- In the section “DASL speakers emphasize the importance of social and interpersonal factors in STEM”, the authors suddenly begin examining keywords across 54 talks, but the number of talks previously stated was 79. Its explained later that only a subset of talks are associated with transcripts, but this should really be made more clear in this section.

Boyle: We added a note in the indicated subheading describing the decrease in N:

We next analyzed themes in DASL talks in more detail by counting mentions of keywords aggregated by topic for each trainee DASL talk with an accessible recording (N = 54).

- When conducting statistical tests, it is generally good practice to list effect sizes immediately alongside the p-value.

Boyle: We confirmed that all p values in the text have effect sizes reported. We added an odds ratio for the participation of women

Overall, women were more likely to participate in DASL than men (odds ratio 1.9, p = 5e-3, one-tailed binomial test).

- Generally, social science literature places the methodology just after the introduction and before the results section. Readers often like to gain a sense of the data before approaching the result. While not strictly necessary, it could be beneficial for the authors to follow this standard.

Boyle: We are more than happy to reorganize the manuscript contents to better meet expectations. We have moved the Methods section after the Introduction as indicated.

- Its confusing that figure 5 is mentioned before figure 4.c,d,e. Either the result text or the panels themselves should be re-arranged

Boyle: We apologize for the mixup. The section discussing the later figure 4 panels has been moved up so that the figures are in order.

- The results of Figure 4d and 4e don’t appear to be mentioned in the discussion section. They deserve to be incorporated into the argument of the paper.

Boyle: We have added a comment on Figure 4d to the Discussion.

Interestingly, speakers pointed primarily to national sources of funding as personally helpful, suggesting a disconnect between the fungible nature of funding and the interpersonal support desired by trainee life scientists (Figure 4d).

Figure 4e is summarized in the first paragraph of the Discussion.

Our tallies of keyword mentions in talk transcripts indicated that social factors like family, education, and mentorship are highly salient and broadly noteworthy (Figure 5a). Sexual and gender minority was noteworthy to fewer life science trainees but still highly salient. Our review of content recommended by speakers addressed offered more evidence for sexual and gender minority identity to select trainees (Figure 4e).

---

## [Decision Letter · Decision Letter 1]

17 Jul 2023

PONE-D-22-17841R1Junior scientists spotlight social bonds in seminars for diversity, equity, and inclusion in STEMPLOS ONE

Dear Dr. Boyle,

Thank you for submitting your manuscript to PLOS ONE. After careful consideration, we feel that it has merit but does not fully meet PLOS ONE’s publication criteria as it currently stands. Therefore, we invite you to submit a revised version of the manuscript that addresses the points raised during the review process.

We look forward to receiving your revised manuscript.

Kind regards,

Claudia Noemi González Brambila, Ph.D.

Academic Editor

PLOS ONE

Journal Requirements:

Reviewers' comments:

Reviewer's Responses to Questions

**Comments to the Author**

1. If the authors have adequately addressed your comments raised in a previous round of review and you feel that this manuscript is now acceptable for publication, you may indicate that here to bypass the “Comments to the Author” section, enter your conflict of interest statement in the “Confidential to Editor” section, and submit your "Accept" recommendation.

Reviewer #1: (No Response)

Reviewer #2: All comments have been addressed

2. Is the manuscript technically sound, and do the data support the conclusions?

Reviewer #1: Yes

Reviewer #2: Yes

3. Has the statistical analysis been performed appropriately and rigorously? 

Reviewer #1: Yes

Reviewer #2: Yes

4. Have the authors made all data underlying the findings in their manuscript fully available?

Reviewer #1: Yes

Reviewer #2: Yes

5. Is the manuscript presented in an intelligible fashion and written in standard English?

Reviewer #1: Yes

Reviewer #2: Yes

6. Review Comments to the Author

Reviewer #1: The authors have largely addressed my concerns, except I still find the motivation around name-based region identification lacking.

The authors have added a self-report survey for validating the name-based tool, which is commendable and strengthens the analysis. However, I still don't understand why they use name-based tools rather than just looking up each of the speakers and manually guessing the region themselves. Is it because it is too hard to guess regions from surnames? Or is it because of discomfort around classifying people demographically by hand?

I find the former reason more defensible than the latter. What I want to prevent is a situation where an algorithmic tool is used to bypass or substitute an ethical consideration. For example, if the authors consider it too problematic to label people by hand, then they shouldn't use an algorithm to do it either! Or rather, if both are fine (which I think it is), and if humans are more accurate, then humans should just do it instead of kicking it off to an algorithm.

All this considered, this is a minor concern, and I don't need to re-review the paper in light of this final suggestion.

Minor comments:

- P8: supplemental figures are referred to like (S1a Fig) and (S1b Fig), while later in P9, it is (S Fig 2). Make labeling scheme consistent. I prefer a scheme like "Fig 1" for main text and "Fig S1" for supplement, so that it is consistent that "Fig" comes first.

- P16: the dashes surrounding "especially Asian men speakers" should be em-dashes

Reviewer #2: I thank the authors for their efforts to improve the manuscript, especially in addressing my concerns regarding the narrative and implications. I believe that the new version of the publication is much stronger and should be published in PLoS One pending only a few exceedingly minor edits.

Minor comments:

- I notice that the authors never actually define the acronym DEI. It is widely used, but it would still be good practice to spell out the full phrase when first used in the manuscript.

- In a similar vein, the authors use the phrase “equity, diversity, and inclusion” in several spots, but also use the acronym DEI. For the sake of consistency, it would be good to always use a consistent ordering, or to always use the acronym.

- Page 3: I appreciate the new methodological paragraph in the introduction that outlines the benefits and drawbacks of programatic inference of speaker identify. However, I feel that it is missing a thesis statement that connects it cleanly with the narrative of the introduction. Perhaps the author could add one sentence in the form: “For these reasons, we choose to use programmatic inference of identify based on each speaker’s name, rather than relying on self-report information.

- Page 5: When mentioning packages like tidy text to SnowballC, the authors should also specify that these are packages in R.

- Page 6: The authors reference a GitHub repository that contains the Wiki2019-LSTM model. It might make sense to also reference the publication with which the repository is associated: https://www.cell.com/cell-systems/pdf/S2405-4712(21)00285-4.pdf

- Page 7: The authors state that manuscripts were uploaded to YouTube to obtain transcripts. To clarify, does that mean that the transcripts are generated using YouTubes’s automatic transcription software? If so, I would make that more explicit in the text, as well as briefly discuss the quality of transcriptions and its limitations.

7. PLOS authors have the option to publish the peer review history of their article (what does this mean?). If published, this will include your full peer review and any attached files.

Reviewer #1: No

Reviewer #2: No

---

## [Author Response · Author response to Decision Letter 1]

19 Sep 2023

Reviewer #1: The authors have largely addressed my concerns, except I still find the motivation around name-based region identification lacking.

We appreciate the reviewer’s contributions to this work and are happy to respond to the remaining concerns below.

The authors have added a self-report survey for validating the name-based tool, which is commendable and strengthens the analysis. However, I still don't understand why they use name-based tools rather than just looking up each of the speakers and manually guessing the region themselves. Is it because it is too hard to guess regions from surnames? Or is it because of discomfort around classifying people demographically by hand?

I find the former reason more defensible than the latter. What I want to prevent is a situation where an algorithmic tool is used to bypass or substitute an ethical consideration. For example, if the authors consider it too problematic to label people by hand, then they shouldn't use an algorithm to do it either! Or rather, if both are fine (which I think it is), and if humans are more accurate, then humans should just do it instead of kicking it off to an algorithm.

We believe that the introduction succinctly summarizes the benefits of computational audits of diversity:

Programmatic and semi-automated methods that make inferences based on names or documents have the potential to curtail response bias, protect participant privacy (e.g., no exposure of phone number, mailing address, or email address), and rescue data lacking self-identification. Indeed, participant name is sufficient for scalable and reproducible (if uneven) inference of race, nationality, gender, and geographic associations [8–12].

To reiterate our findings we added two sentences to the Discussion:

Our semi-automated approach based on the Forebears geospatial names database eliminated errors and irreproducibility due to subjective classification of speaker backgrounds and simplified comparisons across platforms. We also note that the Forebears API can process thousands of surnames near instantaneously with reduced data entry burden.

Having executed this work, the alternative of performing web searches of over 100 individuals and attempting to manually assess their geographic, racial, or ethnic diversity is not an attractive option even if comparable accuracy were attainable. In other settings, such as conferences with thousands of people, it would be prohibitive.

There may be additional information that can be gleaned from a Google search, but in our experience, geographic associations at the granularity presented (e.g. Middle East and North Africa, Hispanic/Latino, or Southeast Asia background) is usually not discernible for junior researchers if databases akin to Forebears are omitted. We are confident that the 4 billion records housed in the Forebears database trumps even expert familiarity with name origins if one were to guess without assistance.

We believe undocumented reasoning and irreproducible results are by far the greatest ethical peril this type of work presents. If subjective judgments were the foundation of the conclusions, the results could easily be skewed (consciously or unconsciously). Our approach is algorithmic, but it is simple to implement and understand. Even if several pages of documentation were included to justify every piece of evidence and analysis in a manual review (e.g. from examining pictures of speakers), future studies or replications would likely differ at least modestly in terms of classification bias.

Minor comments:

- P8: supplemental figures are referred to like (S1a Fig) and (S1b Fig), while later in P9, it is (S Fig 2). Make labeling scheme consistent. I prefer a scheme like "Fig 1" for main text and "Fig S1" for supplement, so that it is consistent that "Fig" comes first.

Good catch. We missed the callout to S Fig 2 when renaming figures to abide by PLOS One guidelines, which declare that Supporting information items be named S[i] Fig or S[i] Table. 

- P16: the dashes surrounding "especially Asian men speakers" should be em-dashes

Another good find. We replaced hyphens and en-dashes with minus signs and em-dashes as needed here and elsewhere throughout the manuscript.

Reviewer #2: I thank the authors for their efforts to improve the manuscript, especially in addressing my concerns regarding the narrative and implications. I believe that the new version of the publication is much stronger and should be published in PLoS One pending only a few exceedingly minor edits.

We’re grateful to the reviewer for offering such detailed feedback in the previous round. We are further enthused by the encouraging remarks on the current draft.

Minor comments:

- I notice that the authors never actually define the acronym DEI. It is widely used, but it would still be good practice to spell out the full phrase when first used in the manuscript.

This was indeed an oversight. The acronym DEI is now specified in the first sentence of the Introduction:

Following spring of 2020, broader recognition of widespread social injustice spurred advocacy for diversity, equity, and inclusion (DEI) in STEM.

- In a similar vein, the authors use the phrase “equity, diversity, and inclusion” in several spots, but also use the acronym DEI. For the sake of consistency, it would be good to always use a consistent ordering, or to always use the acronym.

We agree entirely. All ordering is consistent with “diversity, equity, and inclusion” and most references to diversity, equity, and inclusion are referred to with “DEI” to be more consistent.

- Page 3: I appreciate the new methodological paragraph in the introduction that outlines the benefits and drawbacks of programatic inference of speaker identify. However, I feel that it is missing a thesis statement that connects it cleanly with the narrative of the introduction. Perhaps the author could add one sentence in the form: “For these reasons, we choose to use programmatic inference of identify based on each speaker’s name, rather than relying on self-report information.

We agree that the previous draft did not explicitly connect the listed concerns for automated methods for assessing diversity to the methodology we undertook. We added a sentence to the last paragraph of the Introduction that expresses the sentiment requested: namely that programmatic inference produces a greater volume of usable data that is more reproducible.

Here, we summarize how trainee life scientists conceive of DEI in STEM (Fig 1b). We aggregated gender identity from speaker presentations, algorithmically inferred geography from speaker surnames, and topic mentions gleaned from seminar transcripts to produce highly powered reproducible datasets not limited in scope by preconceived survey questions or speaker response rate in post hoc self-reported information.

- Page 5: When mentioning packages like tidy text to SnowballC, the authors should also specify that these are packages in R.

The methods section now clearly identifies the R packages used. 

Terms in talk titles were counted by parsing titles posted on the DASL website using the R packages tidytext and SnowballC.

Surnames assigned to the Anglophone region were further annotated with race using the predictrace::predict_race command from the predictrace R package. Binomial confidence intervals were obtained using the binconf command from the Hmisc R package.

- Page 6: The authors reference a GitHub repository that contains the Wiki2019-LSTM model. It might make sense to also reference the publication with which the repository is associated: https://www.cell.com/cell-systems/pdf/S2405-4712(21)00285-4.pdf

We appreciate the reviewer pointing this out. The publication is cited in the results section, but now that the methods section is listed before the results section we believe it is essential to cite it there. It is now called out in both places.

We compared our geospatial name inferences to output from a recently published long short-term memory model called Wiki2019-LSTM [8]. The Wiki2019-LSTM model was downloaded from its GitHub repository (https://github.com/greenelab/wiki-nationality-estimate). 

- Page 7: The authors state that manuscripts were uploaded to YouTube to obtain transcripts. To clarify, does that mean that the transcripts are generated using YouTubes’s automatic transcription software? If so, I would make that more explicit in the text, as well as briefly discuss the quality of transcriptions and its limitations.

That is correct: the transcripts are automatically generated by YouTube after uploading the recording. Anecdotally we heard from multiple sources that YouTube’s transcripts were the highest quality free or low-cost commercial option (i.e. without manual review by professionals). We found a reference from Consumer Reports that benchmarked various services and concluded the same.

Publicly viewable DASL and SQUAD talks were downloaded and then uploaded to YouTube, which automatically generated English language captions. Downloaded captions served as talk transcripts. We observed higher fidelity using YouTube than competing automated services such as Zoom, consistent with reports that YouTube produces nearly half as many errors in its audio transcripts: 5 words out of 100 for YouTube versus 8 for Zoom [17]. Transcripts were reviewed and keywords tallied in R.

---

## [Decision Letter · Decision Letter 2]

11 Oct 2023

Junior scientists spotlight social bonds in seminars for diversity, equity, and inclusion in STEM

PONE-D-22-17841R2

Dear Dr. Yeo,

We’re pleased to inform you that your manuscript has been judged scientifically suitable for publication and will be formally accepted for publication once it meets all outstanding technical requirements.

Kind regards,

Claudia Noemi González Brambila, Ph.D.

Academic Editor

PLOS ONE

Additional Editor Comments (optional):

Reviewers' comments:

Reviewer's Responses to Questions

**Comments to the Author**

1. If the authors have adequately addressed your comments raised in a previous round of review and you feel that this manuscript is now acceptable for publication, you may indicate that here to bypass the “Comments to the Author” section, enter your conflict of interest statement in the “Confidential to Editor” section, and submit your "Accept" recommendation.

Reviewer #1: All comments have been addressed

2. Is the manuscript technically sound, and do the data support the conclusions?

Reviewer #1: Partly

3. Has the statistical analysis been performed appropriately and rigorously? 

Reviewer #1: Yes

4. Have the authors made all data underlying the findings in their manuscript fully available?

Reviewer #1: Yes

5. Is the manuscript presented in an intelligible fashion and written in standard English?

Reviewer #1: Yes

6. Review Comments to the Author

Reviewer #1: I have a disagreement but I don't think it's worth holding up the paper's publication for it. I am fine to proceed if the editors are. However, here is my response to the authors:

I disagree with the authors about the utility of human validation for their dataset. For example, there only appear to be three or four speakers with Middle Eastern/North African associated names, and the authors state that none of those people reported demographics. Currently, then, there is no non-algorithmic validation at all that those labels are correct. The type of human-in-the-loop validation I am recommending is that the authors simply check these results for "face validity" and look into whether these labels are plausible.

In fact, 100 people is a tiny dataset, and validating all 100 records by hand is clearly possible, if the authors believed the accuracy of their results to be worth checking.

I sympathize with the authors' complaint that looking up speakers and attempting to discern their geographic background is a difficult and problematic task. However, there are obvious clues that human beings can pick up on, that are employed by sociologists who do qualitative research, that might tell you whether a label has been misplaced. To use an extreme example, if someone writes on their website, "I am from South Asia", but the machine produces a label that says they are from North Africa, then you have an obvious error in the machine labels.

Mostly I bristle at the logical implication of the author's rebuttal, which is that qualitative work is somehow flawed for being "irreproducible", while algorithmic annotation is pure and reproducible simply because one can run code and get the same results back. In fact, this is the same argument used by proponents of predictive policing and courtroom machine learning, since they claim that judges are humans who are biased, but machines can more consistently weigh the available evidence to produce judgments. The issue, of course, is that the machines are themselves biased, and reproduce many of the very structural problems that the authors in starting this speaker series are attempting to address (issues that weren't deeply engaged with by the authors, despite their citations of this literature, such as Lockhart et al.).

However, I think the overall merits of this work outweigh this methodological quibbling, and if one were to take the intersection of my beliefs (that results of this type, especially on datasets as small as this, are generally not trustworthy until humans validate it) and the authors' beliefs (that no human can possibly validate these results), then the paper is just not publishable, since the statistics are insufficient to back up their claims. That is an outcome that I don't think is warranted, largely because I think the outcome of a simple face validity check would confirm that the algorithmic results are indeed good enough. I just want to explicitly call out that both the authors and myself are relying on our priors in making that leap.

7. PLOS authors have the option to publish the peer review history of their article (what does this mean?). If published, this will include your full peer review and any attached files.

Reviewer #1: No

---

## [Editor Report · Acceptance letter]

24 Oct 2023

PONE-D-22-17841R2 

Junior scientists spotlight social bonds in seminars for diversity, equity, and inclusion in STEM 

Dear Dr. Yeo:

I'm pleased to inform you that your manuscript has been deemed suitable for publication in PLOS ONE. Congratulations! Your manuscript is now with our production department. 

Kind regards, 

on behalf of

Dr. Claudia Noemi González Brambila 

Academic Editor

PLOS ONE